# Hazard-Guided Generative Modeling for Sparse and Irreversible Transitions in Longitudinal Disease Trajectories

## Abstract

Understanding longitudinal disease progression is challenging due to sparse and irregular data acquisition. Moreover, clinically meaningful disease-state transitions are rare and irreversible, leading to underrepresentation of critical progression events. Existing generative models often overlook such transitions by treating all time points uniformly, while traditional state-transition models focus on statistical inference rather than data synthesis. In addition, both paradigms often disregard the temporal evolution of patient covariates and labels, treating them as static factors. To address these limitations, we propose a **N**eural **O**DE-based conditional generative model with **HA**zard-guided trajectory sampling (NOHA). By integrating a continuous-time Markov chain with Neural ODE, NOHA effectively models time-varying transition risks and non-linear, irreversible disease dynamics. Our hazard-guided sampling strategy estimates temporal transition hazards to prioritize critical transition events during data synthesis. Moreover, NOHA jointly generates disease trajectories, labels, and patient attributes, ensuring clinically consistent temporal dynamics. Experimental results on four biomarkers from two neurodegenerative disease datasets demonstrate that NOHA generates high-fidelity, progression-aware trajectories that significantly improve downstream disease progression prediction.

## 1. Introduction

Analyzing longitudinal medical data is essential for understanding disease progression, yet modeling such data remains fundamentally challenging due to data scarcity. This primarily stems from the inherently discontinuous, sparse, and time-consuming nature of data acquisition: patient visits are irregular, and follow-up intervals vary widely from months to years (Mueller et al., 2005), which result in inconsistent disease trajectories with heterogeneous and limited observations. Moreover, disease progression in many clinical settings is irreversible (Hampel et al., 2011; Marek et al., 2011), while clinically meaningful state transitions (e.g., from cognitively normal to mild cognitive impairment and to Alzheimer's disease) occur infrequently. When each patient trajectory is treated as a single sample, this leads to severe class imbalance between stable and transitioning cases, causing learning algorithms to be dominated by the stable cases and to underrepresent critical progression events.

To address these issues, deep generative models for time-series data have been developed based on recurrent architectures (Hwang et al., 2019), variational autoencoders (Liu et al., 2023), and diffusion processes (Ho et al., 2020; Cho et al., 2025). Moreover, continuous-time generative models (Grathwohl et al., 2019; Yildiz et al., 2019) based on Neural Ordinary Differential Equations (ODEs) (Chen et al., 2018) and latent ODEs (Rubanova et al., 2019) have enabled modeling under irregular longitudinal sampling. In parallel, classical disease progression modeling methods such as multi-state Markov models (Nicora et al., 2020), continuous-time hidden Markov Models (Wang et al., 2014), and survival models (Lee et al., 2019) explicitly estimate transition hazards of disease states and provide statistical inference of progression dynamics.

However, these lines of work remain fundamentally limited in jointly modeling continuous trajectories and discrete state transitions under irregular longitudinal observations. For example, generative models (Shankar et al., 2023; Song et al., 2025) primarily focus on trajectory forecasting without explicitly modeling the critical disease-state transitions, as they treat all observed time points uniformly. On the other hand, survival- and Markov-based models (Nicora et al., 2020; Meira-Machado et al., 2009) are designed for inference rather than synthesis, without generating realistic data. Moreover, while patient covariates (e.g., age) and diagnostic labels evolve over time, both paradigms typically treat them as static conditions for analyzing disease-related biomarkers, without directly capturing their temporal evolution. As a result, existing methods struggle to capture rare but irre-

[1]Anonymous Institution, Anonymous City, Anonymous Region, Anonymous Country. Correspondence to: Anonymous Author <anon.email@domain.com>.

Preliminary work. Under review by the International Conference on Machine Learning (ICML). Do not distribute.

versible transition events in longitudinal disease trajectories and to faithfully generate complex evolving trajectories.

In this regard, we propose **N**eural **O**DE-based conditional generative model with **HA**zard-guided trajectory sampling (NOHA), a novel generative framework that explicitly characterizes time-varying hazards of label changes in longitudinal data. Here, the hazard represents the instantaneous risk of a state-transition event at time $t$, consistent with the definition in survival analysis (Clark et al., 2003). In this work, NOHA is specifically applied to model the progression of degenerative disorders by capturing non-linear, irreversible disease-state transition risks over time. This is realized by learning non-linear latent disease dynamics via Neural ODEs while simultaneously estimating the temporal hazards of irreversible transitions using a continuous-time Markov chain (CTMC) (Anderson, 2012).

Specifically, we introduce a hazard-guided sampling strategy that estimates piecewise-constant hazards of disease state transitions over each observational interval, naturally accommodating heterogeneous observations across individuals and time windows. The estimated interval-wise hazards are converted into state-transition probabilities, which guide a trajectory generation process to prioritize clinically critical, yet sparse, label-changing intervals. Our model constructs a transition probability distribution over all subjects and time intervals, from which it samples subject–interval pairs where disease state progression is expected. Subsequently, NOHA synthesizes trajectories of disease-related biomarkers that span the sampled interval, using multiple covariates (e.g., age, sex, genotype, data domain) and temporal disease stages as conditions. The synthesized trajectory includes unobserved time points, allowing the model to predict unknown disease dynamics between sparse clinical visits. Moreover, unlike existing generative models that treat covariates and label conditions as fixed inputs, our method jointly generates time-varying covariates and disease labels at the sampled time points along with the biomarkers. This mechanism enables a flexible and progression-aware trajectory synthesis in which biomarkers, patient attributes, and disease labels evolve in a clinically consistent manner.

Our main **contributions** are summarized as follows:

**1)** NOHA performs targeted generation of clinically critical transition events by introducing a novel hazard-guided sampling strategy, which adaptively prioritizes high-risk time points associated with disease state progression.

**2)** NOHA effectively formulates sparse and irreversible disease evolution by combining a CTMC with Neural ODE, naturally handling irregular and non-linear disease dynamics in real-world clinical cohorts.

**3)** NOHA not only uses multiple covariates as conditioning variables for data synthesis, but also jointly generates time-varying covariates and labels over longitudinal trajectories.

As a result, the synthesized trajectories capture richer temporal dynamics with realistic state transitions, leading to improved downstream progression prediction across four biomarkers on two neurodegenerative disease benchmarks.

## 2. Related Works

### 2.1. Generative Models for Longitudinal Data

Unlike cross-sectional data collected from a single time point, longitudinal data exhibit complex temporal dependencies and irregular sampling intervals. Generative modeling of such data aims to characterize the underlying distribution of temporal sequences. Early studies primarily utilized Recurrent Neural Networks (RNNs) (Lipton et al., 2016) and Long Short-Term Memory (LSTM) (Zhang et al., 2019) to capture sequential dependencies. More recently, Variational Autoencoders (VAEs) (Fortuin et al., 2020) and normalizing flows (Hwang et al., 2019) have been explored to map longitudinal trajectories into structured latent spaces, while Generative Adversarial Networks (GANs) (Yoon et al., 2019) and diffusion models (Cho et al., 2025; Tashiro et al., 2021) have been adopted to improve the realism of generated sequences. While these methods often struggle with the irregular sampling intervals, Neural ODEs (Rubanova et al., 2019) naturally handle such irregularity by modeling the evolution of hidden states as a continuous-time function, making them suitable for longitudinal data generation.

### 2.2. State-Transition Modeling and Survival Analysis

While generative models aim to capture overall trajectories, state-transition modeling and survival analysis provide a formal framework for characterizing discrete status changes and event timing. State-transition approaches are typically categorized into discrete and continuous-time Markov chains (Ross, 1995), i.e., DTMC and CTMC. While DTMC assumes that transitions occur at fixed intervals, CTMC allows transitions at arbitrary time points, making it suited for capturing the irregularity of longitudinal data. Within this continuous-time framework, survival analysis (Cox, 1972; Aalen et al., 2008) is viewed as a special case that models a single transition from an initial state to a terminal event state. This is further extended to multi-state models (Meira-Machado et al., 2009; Le-Rademacher et al., 2022), which consider multiple intermediate states. Building upon these frameworks, we introduce a joint modeling method that integrates multi-state transitions with a Neural ODE, enabling adaptive estimation of individual event risks while robustly handling the continuous evolution of longitudinal disease dynamics.

## 3. Method

In this section, we describe our **N**eural **O**DE-based conditional generative model with **HA**zard-guided trajectory

**Figure 1. Overview of the model architecture.** Using a sequence of labels $\boldsymbol{y}_i$ and covariates $\boldsymbol{\Gamma}_i$ as conditions, a sequence of data $\boldsymbol{x}_i$ with $K_i$ time points is embedded into a latent space as $\boldsymbol{z}_i = \{z_{i,k}\}_{k=1}^{K_i}$. Neural ODE $f_\theta$ learns the sequential disease dynamics by reconstructing the observed trajectory as $\hat{\boldsymbol{x}}_i$. In parallel, a hazard network $g_\phi$ estimates the transition hazard $q_{i,k}^{c \to c+1}$ for disease stage progression under a CTMC framework, where only monotonic transitions from a label $c$ to $c+1$ are allowed. The estimated hazards are converted into transition probabilities $p_{i,k}^{c \to c+1}$ and used to guide trajectory sampling, prioritizing underrepresented progression intervals involving state transitions. Finally, the original and augmented datasets are jointly used to train a conditional longitudinal classifier, improving disease stage prediction under limited and imbalanced longitudinal data setting.

sampling (NOHA). We first define the problem setting and introduce our conditional Neural ODE for modeling temporal data and the hazard-guided trajectory sampling strategy, which constitute the core components of our method. Following the generation phase, we explain how the synthesized trajectories are leveraged for longitudinal disease stage classification in an end-to-end scheme. The overall model architecture is illustrated in Fig. 1, and we further provide pseudo-code of the model architecture in Appendix A.

### 3.1. Problem Definition: Generating Longitudinal Data with Sparse and Irreversible Transitions

Let $\boldsymbol{x}_i = \{x_{i,k}\}_{k=1}^{K_i}$ denote a longitudinal sample of subject $i$ with $K_i$ time points, where each $x_{i,k} \in \mathbb{R}^F$ is a set of independent features (e.g., brain measurements from $F$ brain regions of interests) observed at age $a_{i,k}$. Each subject has an ordinal label sequence $\boldsymbol{y}_i = \{y_{i,k}\}_{k=1}^{K_i}$, where $y_{i,k} \in \{1, \ldots, C\}$ and $y_{i,k} \leq y_{i,k+1}$, reflecting the irreversible progression of degenerative disease severity. We further define a covariate set $\boldsymbol{\Gamma}_i = \boldsymbol{\gamma}_i \cup \bar{\boldsymbol{\gamma}}_i$, which consists of a set of time-varying covariates $\boldsymbol{\gamma}_i$ (e.g., age) and a set of time-invariant covariates $\bar{\boldsymbol{\gamma}}_i$ (e.g., sex, genotype, domain).

Given a population of longitudinal training samples $\mathcal{D} = \{\boldsymbol{x}_i, \boldsymbol{y}_i, \boldsymbol{\Gamma}_i\}_{i=1}^N$, where *only a limited number of subjects exhibit actual disease stage transitions over time*, our goal is to develop a continuous-time generative model that jointly captures the temporal disease dynamics and adaptively synthesizes longitudinal trajectories with disease stage changes. Specifically, our model synthesizes time-varying variables $\boldsymbol{x}_i, \boldsymbol{y}_i$, and $\boldsymbol{\gamma}_i$, while treating the time-invariant covariates $\bar{\boldsymbol{\gamma}}_i$ as deterministic conditioning variables.

To explicitly model their temporal patterns, we propose a Neural ODE-based framework that naturally handles sparse and irregular observation times and exploits multiple covariates ($\boldsymbol{y}_i$ and $\boldsymbol{\Gamma}_i$) as conditions for generating $\boldsymbol{x}_i$, enabling

realistic disease trajectory generation in practical clinical settings. Moreover, our model learns sequential hazards of ordinal disease stage transitions over time using a CTMC. By using the transition hazards, our model adaptively augments underrepresented, stage-changing longitudinal samples with high transition hazards. Following the generation phase, the synthesized trajectories are exploited to improve a downstream longitudinal disease stage classification in an end-to-end manner to demonstrate their practical utility.

### 3.2. Modeling Longitudinal Dynamics via Neural ODE

To characterize the continuous-time evolution of disease trajectories, the baseline observation at the initial time point ($k = 1$) is first embedded into a latent space using an encoder $\text{Enc}(\cdot)$ (i.e., MLP with ReLU) as

$$z_i(a_{i,1}) = \text{Enc}\big([x_{i,1}, y_{i,1}, \boldsymbol{\gamma}_{i,1}, \bar{\boldsymbol{\gamma}}_i]\big), \qquad (1)$$

yielding an initial latent representation $z_i$ at age $a_{i,1}$. Given an arbitrary time $t$ ($t > a_{i,1}$), a Neural ODE $\frac{dz_i(t)}{dt} = f_\theta\big(z_i(t), t\big)$ computes the continuous-time latent trajectory up to $t$ from the initial representation $z_i(a_{i,1})$ as follows

$$
\begin{aligned}
z_i(t) &= \text{ODESolve}\big(z_i(a_{i,1}), f_\theta, a_{i,1}, t\big) \\
&= z_i(a_{i,1}) + \int_{a_{i,1}}^{t} f_\theta\big(z_i(\tau), \tau\big)\, d\tau.
\end{aligned} \qquad (2)
$$

To learn the temporal dynamics parameterized by $f_\theta$, the continuous latent trajectory in Eq. (2) is decoded at the observed ages $\{a_{i,k}\}_{k=1}^{K_i}$. Specifically, the latent states $z_{i,k} = z_i(a_{i,k})$ are mapped back to the observation space using a decoder $\text{Dec}(\cdot)$ (i.e., MLP with ReLU) as

$$\hat{x}_{i,k} = \text{Dec}\big([z_{i,k}, y_{i,k}, \boldsymbol{\gamma}_{i,k}, \bar{\boldsymbol{\gamma}}_i]\big), \qquad (3)$$

and evaluated with the following $l2$ reconstruction loss

$$L_{\text{rec}} = \mathbb{E}_{i,k}\big[\|\hat{x}_{i,k} - x_{i,k}\|_2^2\big]. \qquad (4)$$

Minimizing $L_{\text{rec}}$ trains $f_\theta$ to capture smooth, continuous-time trajectories from sparsely sampled longitudinal data.

### 3.3. Learning the Hazard of Disease Stage Transition

To model the irreversible progression of degenerative diseases in sparsely sampled longitudinal data, we first define a piecewise-constant risk of the transition event at a specific time interval as a hazard, and compute temporal hazards of label transitions using a CTMC. Specifically, we consider a $C$-state CTMC in which only monotonic transitions from stage $c$ to $c+1$ are allowed ($c = 1, \ldots, C$).

The temporal dynamics of the CTMC are fully characterized by a transition rate matrix (a.k.a. infinitesimal generator), which specifies the instantaneous transition rates between states. Since longitudinal clinical measurements are generally sparsely and irregularly sampled, we model the CTMC with piecewise-constant hazards over each observation interval $[k, k+1]$, which provides a data-efficient approximation while retaining the continuous-time formulation. Accordingly, we define an interval-specific generator matrix $\mathbf{Q}_{i,k}$:

$$\mathbf{Q}_{i,k} = \begin{pmatrix} -q_{i,k}^{1\to2} & q_{i,k}^{1\to2} & 0 & \cdots & 0 \\ 0 & -q_{i,k}^{2\to3} & q_{i,k}^{2\to3} & \cdots & 0 \\ 0 & 0 & -q_{i,k}^{3\to4} & \cdots & \vdots \\ \vdots & \vdots & \ddots & \ddots & q_{i,k}^{C-1\to C} \\ 0 & 0 & \cdots & 0 & 0 \end{pmatrix}, \quad (5)$$

whose off-diagonal entries $q_{i,k}^{c\to c+1} \geq 0$ represent the hazard rate of transitioning from a stage $c$ to the next stage $c+1$ over the interval $[k, k+1]$.

Since the practical interval length is heterogeneous across individuals and time points, we set the interval length to be a time difference $\Delta a_{i,k} = a_{i,k+1} - a_{i,k}$ and employ these two age endpoints as conditions in deriving the transition hazard rate. Accordingly, a set of transition hazard rates $\{q_{i,k}^{c\to c+1}\}_{c=1}^{C-1}$ is parameterized by a hazard network $g_\phi$ (i.e., MLP with ReLU) and is enforced to be non-negative via a softplus function $\mu(\cdot)$ as

$$\{q_{i,k}^{c\to c+1}\}_{c=1}^{C-1} = \mu\big(g_\phi\big([z_{i,k}, y_{i,k}, a_{i,k}, a_{i,k+1}]\big)\big). \quad (6)$$

Given the $\Delta a_{i,k}$, the state transition probability $p_{i,k}^{c\to c+1}(\Delta a_{i,k})$ within this interval is defined under the CTMC assumption (Norris, 1998) as follows:

$$p_{i,k}^{c\to c+1}(\Delta a_{i,k}) = 1 - \exp\big(-q_{i,k}^{c\to c+1} \Delta a_{i,k}\big) \in [0, 1]. \quad (7)$$

For each interval, we define a binary indicator $y'_{i,k} = \mathbf{1}[y_{i,k} = c \wedge y_{i,k+1} = c+1]$ that specifies whether an actual transition from $c$ to $c+1$ is observed. Treating each interval as a Bernoulli event (i.e., whether a stage transition occurs or not), we optimize a negative log-likelihood using a binary cross-entropy loss $\ell_{i,k}$ defined as follows:

$$\ell_{i,k} = -\big(y'_{i,k} \log p_{i,k}^{c\to c+1} + (1-y'_{i,k}) \log(1-p_{i,k}^{c\to c+1})\big). \quad (8)$$

To exclude invalid intervals (e.g., terminal stage $C$ or the last observation at $K_i$), we apply a mask $m_{i,k} = \mathbf{1}[y_{i,k} \in \{1, \ldots C-1\}] \cdot \mathbf{1}[k < K_i]$ and compute a hazard loss as

$$L_{\text{hazard}} = \mathbb{E}_{i,k}\big[m_{i,k} \cdot \ell_{i,k}\big], \quad (9)$$

which trains the hazard network $g_\phi^c$ to capture time-dependent, stage-specific progression risks.

### 3.4. Hazard-Guided Trajectory Sampling

To augment underrepresented, state-deteriorating longitudinal trajectories, we propose a hazard-guided trajectory sampling strategy that preferentially selects time intervals with a high probability of disease stage transition. Specifically, we first define a categorical distribution over subject–interval pairs $(i, k)$ using the $p_{i,k}^{c\to c+1}$ in Eq. (7) as

$$(i, k) \sim \Pr(i, k) = \frac{m_{i,k} \cdot p_{i,k}^{c\to c+1}}{\sum_{i'} \sum_{k'} m_{i',k'} \cdot p_{i',k'}^{c\to c+1}} \quad (10)$$

which prioritizes intervals $[k, k+1]$ with higher transition risk while excluding invalid segments via the mask $m_{i,k}$. This sampling scheme explicitly focuses on rare but clinically meaningful progression events, thereby counteracting the severe imbalance in longitudinal disease trajectories.

Given a sampled pair $(i, k)$, we introduce how to sample a sequence of time-varying variables (i.e., $\boldsymbol{\gamma}_i$, $\boldsymbol{y}_i$, $\boldsymbol{x}_i$) that spans the target interval $[k, k+1]$ of subject $i$ in the following paragraphs. Specifically, we select a sequence of time points spanning the target interval and assign corresponding labels at the sampled time points. We then propagate the latent states across the selected time points using the $f_\theta$ and decode them back to the observation space, yielding hazard-guided synthetic trajectories that are temporally consistent and enriched with plausible stage transitions.

**Sampling of time-varying covariates.** We first set a random sequence length $\tilde{K}_i \in [2, K_i^{\max}]$, where $K_i^{\max}$ is the maximum sequence length observed in the training set. To sample a sequence of pseudo-time points $\tilde{k} = 1, \ldots, \tilde{K}_i$ that spans the target interval $[k, k+1]$, we use age as the continuous time variable, such that the subject's age represents the time of each observation.

*Remark.* Note that in our experimental scenario, the time-varying covariate $\boldsymbol{\gamma}_i$ only include age, as it is the only variable consistently available across subjects in a time-resolved manner. In other words, we select pseudo-ages $\{\tilde{a}_{i,\tilde{k}}\}_{\tilde{k}=1}^{\tilde{K}_i}$ based on the observed ages $a_{i,k}$ and $a_{i,k+1}$ at the target interval, where the first and last pseudo-ages are chosen before and after the interval, respectively, as follows:

$$\tilde{a}_{i,1} \in \{a_{i,j}\}_{j=0}^{k}, \quad \text{and} \quad \tilde{a}_{i,\tilde{K}_i} \in \{a_{i,j}\}_{j=k+1}^{K_i}. \quad (11)$$

Here, we define the selected starting time point as $\tilde{a}_{i,1} = a_{i,\text{start}}$. We then randomly sample $\tilde{K}_i - 2$ intermediate ages

between them as $\tilde{a}_{i,2:\tilde{K}_i-1} \sim \text{Uniform}(\tilde{a}_{i,1}, \tilde{a}_{i,\tilde{K}_i})$, ensuring a strictly increasing order, $\tilde{a}_{i,\tilde{k}} < \tilde{a}_{i,\tilde{k}+1}$.

**Sampling of labels.** To assign pseudo-labels $\{\tilde{y}_{i,\tilde{k}}\}_{\tilde{k}=1}^{\tilde{K}_i}$ at the pseudo-time points, we perform sequential sampling from the first time point $\tilde{k} = 1$, by using transition hazards and probabilities predicted by the hazard network $g_\phi$. Given the initial label $\tilde{y}_{i,1} = y_{i,\text{start}}$ and latent feature $\tilde{z}_i(\tilde{a}_{i,1}) = z(a_{i,\text{start}})$ driven by Eq. (1), a hazard network $g_\phi$ yields CTMC-based hazard rates as

$$\{\tilde{q}_{i,\tilde{k}}^{c \to c+1}\}_{c=1}^{C-1} = \mu\Big(g_\phi\big([\tilde{z}_{i,\tilde{k}}, \tilde{y}_{i,\tilde{k}}, \tilde{a}_{i,\tilde{k}}, \tilde{a}_{i,\tilde{k}+1}]\big)\Big), \quad (12)$$

as in Eq. (6). Similarly, we then convert the hazard rate into a transition probability as in Eq. (7):

$$\tilde{p}_{i,\tilde{k}}^{c \to c+1}(\Delta\tilde{a}_{i,\tilde{k}}) = 1 - \exp\big(-\tilde{q}_{i,\tilde{k}}^{c \to c+1}\Delta\tilde{a}_{i,\tilde{k}}\big), \quad (13)$$

where $\Delta\tilde{a}_{i,\tilde{k}} = \tilde{a}_{i,\tilde{k}+1} - \tilde{a}_{i,\tilde{k}}$. Finally, we sample the next pseudo-label autoregressively:

$$\tilde{y}_{i,\tilde{k}+1} = \begin{cases} \tilde{y}_{i,\tilde{k}}, & \text{if } (u \geq \tilde{p}_{i,\tilde{k}}^{c \to c+1}) \wedge (\tilde{y}_{i,\tilde{k}} < C), \\ \tilde{y}_{i,\tilde{k}} + 1, & \text{if } (u < \tilde{p}_{i,\tilde{k}}^{c \to c+1}) \wedge (\tilde{y}_{i,\tilde{k}} < C), \\ C, & \text{if } (\tilde{y}_{i,\tilde{k}} = C), \end{cases}$$

$$(14)$$

where $u \sim \text{Uniform}(0,1)$. This procedure enforces monotonic disease progression and assumes an absorbing state $C$ as the maximally progressed state. The resulting pseudo-label sequence $\{\tilde{y}_{i,\tilde{k}}\}_{\tilde{k}=1}^{\tilde{K}_i}$ is consistent with the hazard model and flexibly adapts to a plausible disease progression over the augmented irregular time grid.

**Sampling of disease trajectories.** Given the initial latent state $\tilde{z}_i(\tilde{a}_{i,1}) = z(a_{i,\text{start}})$, we use Neural ODE $f_\theta$ to obtain the latent trajectory over the pseudo-times as in Eq. (2):

$$\tilde{z}_{i,\tilde{k}}(t) = \tilde{z}_{i,\tilde{k}}(\tilde{a}_{i,\tilde{k}}) + \int_{\tilde{a}_{i,\tilde{k}}}^{t} f_\theta\big(\tilde{z}_{i,\tilde{k}}(\tau), \tau\big)\, d\tau, \quad (15)$$

where the latent state at each pseudo-time is obtained by evaluating the trajectory at $t = \tilde{a}_{i,\tilde{k}}$, i.e., $\tilde{z}_{i,\tilde{k}} = \tilde{z}_{i,\tilde{k}}(t)\big|_{t=\tilde{a}_{i,\tilde{k}}}$. We then decode each latent feature back to the observation space:

$$\tilde{x}_{i,\tilde{k}} = \text{Dec}\big([\tilde{z}_{i,\tilde{k}}, \tilde{y}_{i,\tilde{k}}, \tilde{\boldsymbol{\gamma}}_{i,\tilde{k}}, \bar{\boldsymbol{\gamma}}_i]\big) \quad (16)$$

as in Eq. (3), where a set of time-varying covariates $\tilde{\boldsymbol{\gamma}}_i$ includes the augmented ages $\tilde{a}_{i,\tilde{k}}$. This process produces hazard-guided synthetic trajectories that are temporally consistent and enriched with plausible stage transitions.

### 3.5. Longitudinal Disease Stage Classification

We construct the augmented dataset $\mathcal{D}_{\text{aug}}$ that consists of synthesized longitudinal data $\tilde{x}_i = \{\tilde{x}_{i,\tilde{k}}\}_{\tilde{k}=1}^{\tilde{K}_i}$, pseudo-labels $\tilde{\boldsymbol{y}}_i = \{\tilde{y}_{i,\tilde{k}}\}_{\tilde{k}=1}^{\tilde{K}_i}$, and covariates $\tilde{\boldsymbol{\Gamma}}_i = \tilde{\boldsymbol{\gamma}}_i \cup \bar{\boldsymbol{\gamma}}_i$. On the

combined dataset $\mathcal{D} \cup \mathcal{D}_{\text{aug}}$, a conditional autoregressive classifier (e.g., RNN) is trained to model the longitudinal stage prediction

$$p(\hat{y}_{i,k} \mid x_{i,\leq k}, y_{i,<k}, \boldsymbol{\gamma}_{i,\leq k}, \bar{\boldsymbol{\gamma}}_i), \quad (17)$$

where the model predicts a disease stage $\hat{y}_{i,k}$ sequentially over time, conditioned on data, covariates, and past labels. The classifier is optimized using a cross-entropy loss:

$$L_{\text{cls}} = \mathbb{E}_{i,k}\big[\text{CE}(\hat{y}_{i,k}, y_{i,k})\big]. \quad (18)$$

By jointly training on both real and hazard-guided synthetic trajectories, the classifier learns more robust and temporally consistent disease stage predictors under limited and imbalanced longitudinal data.

## 4. Experiments

In this section, we present the quantitative superiority of NOHA over nine baseline methods on four independent experiments using four biomarkers provided from two longitudinal, neurodegenerative Alzheimer's disease (AD) benchmarks: Alzheimer's Disease Neuroimaging Initiative (ADNI) (Mueller et al., 2005) and Open Access Series of Imaging Studies (OASIS) (LaMontagne et al., 2019), whose demographics are presented in Appendix B. Moreover, we discuss the effects of model components, along with the results of ablation studies. Due to the page limit, detailed implementation setups are reported in Appendix C.

### 4.1. Datasets

**ADNI.** ADNI offers the largest publicly available longitudinal AD dataset, including rich patient data such as age, sex, genotypes, and multimodal imaging biomarkers. We used four AD-related biomarkers collected from MRI and PET: (1) cortical thickness (CT) from MRI, Standardized Uptake Value Ratio (SUVR) of (2) Amyloid, (3) Fluorodeoxyglucose (FDG), and (4) Tau from PET. All biomarkers were measured on $F = 148$ brain regions of interest (ROIs) based on the Destrieux atlas (Destrieux et al., 2010).

We excluded subjects with a single time point in the ADNI, which resulted in $N=102, 478, 642, 160$ subjects for CT, Amyloid, FDG, and Tau, respectively. The number of time points per subject, $K_i$, ranges from 2 to 10, and the observation intervals, $\Delta a_{i,k}$, range from 3 months to 8 years. At each time point, subjects are diagnosed into one of three states: Cognitively Normal (CN), Mild Cognitive Impairment (MCI), and AD, where disease progression is irreversible from CN to AD. Given the $N$ trajectories, let the numbers of subjects with and without label transitions be $N_{\text{trans}}$ and $N_{\text{w/o trans}}$, respectively, where $N_{\text{trans}} + N_{\text{w/o trans}} = N$. For each biomarker, the $N_{\text{trans}}$ and $N_{\text{w/o trans}}$ are 16/86 for CT, 104/374 for Amyloid, 156/486 for FDG, and 6/154 for Tau, respectively, showing a severe imbalance.

**OASIS.** OASIS provides a relatively small longitudinal dataset in which only Tau SUVR is available for $N = 32$ subjects, each with two time points separated by a 6-month interval. Similar to ADNI, Tau SUVR was extracted from $F = 148$ ROIs based on the Destrieux atlas at each time point. All subjects were diagnosed as either CN or AD; only 2 subjects exhibited a label transition from CN to AD, while the remaining 30 subjects maintained a consistent diagnosis. Due to the limited dataset size, we performed multi-domain learning by jointly using OASIS and ADNI in the Tau experiment to improve generalizability, where domain information was used as a proxy for genotype features since OASIS does not provide genotype data.

### 4.2. Experimental Setup

**Baseline Methods.** Along with the setting without data augmentation (denoted as 'No Aug.' in Tables 1 and 2), we adopt nine baseline methods, including a traditional interpolation-based data synthetic technique and various generative models such as VAEs, normalizing flow models, GANs, and diffusion methods. Specifically, SMOTE (Chawla et al., 2002) is an interpolation-based method that synthesizes data points as convex combinations of a real sample and its $k$-nearest neighbors. Despite its simplicity, SMOTE has demonstrated its competitive performance in various prior studies (Camino et al., 2020; Cho et al., 2025). For VAE-based methods, we adopt TVAE (Kingma, 2014), a VAE tailored for tabular data, and GOGGLE (Liu et al., 2023), which incorporates graph learning to model feature dependencies. For a normalizing-flow model, we use CRow (Hwang et al., 2019), which generates medical sequential data under conditional settings. We further include GAN-based approaches, including TimeGAN (Yoon et al., 2019) for time-series generation, CTAB-GAN (Zhao et al., 2021), and its improved variant CTAB-GAN+ (Zhao et al., 2024). Lastly, diffusion-based methods include DDPM (Ho et al., 2020) and Con-DOR (Cho et al., 2025), a conditional diffusion model designed for longitudinal medical data synthesis. All generative models were trained in a conditional scheme, where patients' sex, genotype, age, data domain, and diagnostic labels were provided as conditioning variables.

**Evaluation.** All methods generate $N_{\text{aug}} = N_{\text{w/o trans}} - N_{\text{trans}}$ samples per biomarker, which are used as additional training data for downstream sequence classification. To assess the generalizability of augmented data, we used two independent classifiers: a conditional RNN (Cho et al., 2014) and a conditional autoregressive Transformer (Vaswani et al., 2017). All data were split into train and test sets with an 8:2 ratio, and all models were trained three times with different parameter initializations. Mean test performance of sequence classification is reported in Tables 1 and 2, and generation time is compared in Table 3.

We evaluated longitudinal data using two metrics and additionally used five metrics for cross-sectional evaluation by treating each time point as an independent entity. For longitudinal evaluation, we computed trajectory-level mean accuracy across all subject-wise time points $K_i$ as follows:

$$\frac{1}{|\mathcal{D}_{\text{test}}|} \sum_{i=1}^{|\mathcal{D}_{\text{test}}|} \frac{1}{K_i} \sum_{k=1}^{K_i} \mathbf{1}(y_{i,k} = \hat{y}_{i,k}), \tag{19}$$

where $\mathcal{D}_{\text{test}}$ denotes the test set, and $\mathbf{1}(\cdot)$ is the indicator function. To explicitly assess prediction performance on disease progression events, we additionally report *transition-only trajectory accuracy*, which is computed in the same manner as Eq. (19) but restricted to test samples that contain at least one label transition. For cross-sectional evaluation, we adopt accuracy, F1-score, precision, recall, and specificity.

### 4.3. Quantitative Results

#### 4.3.1. CLASSIFICATION PERFORMANCE COMPARISON

Tables 1 and 2 show the downstream disease progression classification performance using a conditional RNN and a conditional autoregressive Transformer, respectively. For classification, the synthetic trajectories generated by each method were combined with the original training set as additional training data. Across all biomarkers and classifiers, NOHA consistently outperforms baseline methods or achieves the second-best performance in longitudinal evaluation metrics. In particular, NOHA surpasses all baselines in the transition-only trajectory accuracy (the second column), which exclusively evaluates trajectories with label transitions. These results demonstrate that our transition-guided learning strategy effectively captures clinically critical disease progression events that are otherwise sparse and difficult to model. In cross-sectional evaluations, where each time point is treated as an independent sample, NOHA achieves the highest F1-score and the best or second-best accuracy and recall across all settings. These results suggest that the generated samples effectively reduce false-negative predictions and enhance sensitivity to disease progression. This indicates that NOHA improves not only trajectory-level modeling but also the quality and diversity of point-wise feature distributions for cross-sectional classification.

#### 4.3.2. COMPARISON ON GENERATION TIME

Table 3 reports the computational cost for generating $N_{\text{aug}} = 210$ sequences in the amyloid experiment. Specifically, SMOTE achieves the fastest generation time due to its simple interpolation-based mechanism, whereas deep generative models such as GOGGLE, CTAB-GAN, and DDPM require substantially longer runtimes owing to computationally intensive and iterative sampling schemes. In contrast, our method outperforms all deep generative baselines in efficiency, requiring only 0.491 seconds, which is 188 times

*Table 1.* Classification performance using an RNN. The average of three replicates and their standard deviations are reported. The best results are in bold, and the second-best are underlined.

| Methods | Longitudinal evaluation | | Cross-sectional evaluation | | | | |
|---|---|---|---|---|---|---|---|
| | Accuracy | | Accuracy | F1 | Precision | Recall | Specificity |
| | All | Transition-only | | | | | |
| **Cortical Thickness** | | | | | | | |
| No Aug. | 65.9±9.6 | 50.0±28.9 | 64.1±7.9 | 58.0±3.0 | 65.5±1.0 | 62.9±6.2 | 79.3±3.8 |
| SMOTE | **74.6±7.3** | 44.4±19.3 | **73.9±6.3** | 63.0±13.8 | 67.3±19.4 | 62.2±12.1 | 83.9±4.0 |
| TVAE | 64.3±6.3 | 61.1±19.2 | 65.4±7.4 | 61.9±2.2 | 72.6±8.8 | 63.3±7.2 | 79.1±3.5 |
| GOGGLE | 63.0±10.0 | 57.4±8.5 | 63.4±9.3 | 62.2±6.5 | 73.8±7.0 | 64.7±13.0 | 77.8±5.9 |
| CRow | 60.3±1.4 | 55.6±19.3 | 61.4±4.5 | 59.0±7.8 | 63.0±7.8 | 67.5±13.0 | 78.3±3.1 |
| TimeGAN | 63.4±2.5 | 33.3±0.0 | 64.1±3.0 | 54.7±2.1 | 62.0±7.2 | 54.7±1.9 | 77.8±2.0 |
| CTAB-GAN | 60.3±7.3 | 33.3±0.0 | 61.4±5.7 | 53.0±5.3 | 59.2±5.4 | 52.3±4.6 | 75.5±4.0 |
| CTAB-GAN+ | 60.9±4.7 | 48.1±12.8 | 60.8±5.2 | 57.2±5.1 | 62.1±8.7 | 63.3±10.2 | 77.3±3.0 |
| DDPM | 69.1±6.4 | 59.3±12.8 | 69.3±6.9 | 60.6±1.6 | 64.1±1.8 | 69.6±5.6 | **84.0±3.6** |
| ConDOR | 65.1±3.6 | 61.1±9.6 | 63.4±3.0 | 62.3±1.5 | 77.6±8.1 | 66.0±8.5 | 78.8±1.6 |
| NOHA (Ours) | 73.0±2.8 | **66.7±0.0** | 71.9±2.3 | **73.1±1.0** | **82.7±6.5** | **75.0±7.0** | 83.8±1.9 |
| **Amyloid** | | | | | | | |
| No Aug. | 57.3±2.8 | 52.0±6.3 | 56.4±2.5 | 49.8±7.1 | 65.4±4.0 | 49.1±5.8 | 73.1±2.0 |
| SMOTE | 56.6±1.2 | 57.7±3.5 | 56.1±1.2 | 53.6±3.1 | 63.0±4.4 | 52.5±2.7 | 73.3±1.1 |
| TVAE | 56.8±2.8 | 60.3±2.1 | 57.4±2.6 | 49.2±4.9 | 66.8±1.0 | 50.9±4.6 | 73.5±2.0 |
| GOGGLE | 53.8±1.6 | 50.6±12.5 | 53.8±1.4 | 45.3±4.4 | 65.6±8.0 | 44.9±3.6 | 71.7±0.6 |
| CRow | 53.2±3.7 | 50.4±8.6 | 53.7±3.6 | 46.7±12.4 | 70.3±11.8 | 49.3±9.3 | 74.1±4.3 |
| TimeGAN | 53.5±1.4 | 52.5±6.5 | 53.0±0.6 | 48.0±6.8 | 66.5±5.1 | 49.0±6.2 | 73.7±2.5 |
| CTAB-GAN | 54.8±3.1 | 58.7±2.6 | 55.3±2.0 | 50.1±6.0 | 61.3±8.1 | 50.8±2.7 | 73.4±0.7 |
| CTAB-GAN+ | 54.7±1.0 | 61.2±3.0 | 54.4±1.9 | 47.4±4.1 | 66.0±6.6 | 49.9±3.2 | 72.4±0.7 |
| DDPM | 57.0±1.7 | 49.0±9.5 | 57.2±1.8 | 47.3±5.0 | 71.3±1.8 | 47.9±5.1 | 73.2±1.2 |
| ConDOR | 59.1±1.5 | 61.1±1.9 | 58.3±0.7 | 52.6±4.5 | 69.6±6.7 | 54.6±2.0 | 74.2±0.9 |
| NOHA (Ours) | **61.9±0.9** | **69.2±3.3** | **61.8±0.9** | **57.1±1.1** | 68.9±5.1 | **57.1±2.4** | **76.7±1.0** |
| **FDG** | | | | | | | |
| No Aug. | 50.7±2.0 | 42.4±10.3 | 52.7±1.4 | 46.6±7.0 | 62.5±5.5 | 50.0±7.0 | 74.5±2.9 |
| SMOTE | 48.2±2.6 | 42.2±1.6 | 52.0±1.3 | 47.4±3.7 | 60.0±6.7 | 50.1±4.2 | 74.7±1.4 |
| TVAE | 55.8±0.5 | 51.2±3.9 | 57.2±0.2 | 56.1±1.0 | 60.4±1.8 | 55.0±1.3 | 76.5±0.5 |
| GOGGLE | 49.9±1.8 | 50.6±5.2 | 50.1±1.5 | 36.6±3.2 | 65.2±3.9 | 40.7±2.8 | 70.4±1.8 |
| CRow | 59.0±3.1 | 36.7±3.2 | 63.6±1.9 | 51.8±2.6 | 67.0±9.4 | 57.0±1.8 | 80.6±0.9 |
| TimeGAN | 61.9±3.1 | 38.5±1.9 | 65.7±2.7 | 58.5±6.3 | 67.7±3.3 | 60.6±3.6 | 81.8±1.3 |
| CTAB-GAN | 49.0±1.8 | 46.9±11.9 | 49.4±1.5 | 35.5±7.9 | 69.7±9.0 | 40.3±4.0 | 70.1±1.3 |
| CTAB-GAN+ | 48.9±4.3 | 39.8±7.8 | 49.4±6.1 | 39.7±3.1 | 64.4±3.2 | 47.5±2.6 | 73.3±0.3 |
| DDPM | 52.4±0.3 | 43.3±0.5 | 54.1±0.3 | 51.5±0.9 | 63.4±1.8 | 52.4±1.5 | 75.6±0.7 |
| ConDOR | 56.0±4.6 | 49.6±5.9 | 57.6±4.1 | 54.8±8.0 | 62.2±3.6 | 56.7±5.7 | 77.6±2.0 |
| NOHA (Ours) | 61.5±1.6 | **55.4±2.2** | 61.5±2.4 | 60.9±3.0 | 63.2±3.0 | 60.5±3.3 | 79.0±1.9 |
| **Tau (ADNI + OASIS)** | | | | | | | |
| No Aug. | 58.9±3.5 | 34.7±9.6 | 57.6±1.9 | 40.1±2.8 | 45.1±1.9 | 45.4±2.1 | 75.8±1.8 |
| SMOTE | 52.1±1.8 | 44.4±0.0 | 52.1±2.6 | 40.0±0.4 | 39.5±2.2 | 41.8±1.0 | 72.4±1.0 |
| TVAE | 55.7±1.5 | 45.8±0.0 | 55.1±1.7 | 42.2±4.6 | 50.6±3.1 | 44.9±1.7 | 74.2±0.8 |
| GOGGLE | 54.5±2.0 | 51.4±4.8 | 55.1±1.7 | 42.6±2.6 | 48.1±6.1 | 44.8±1.7 | 74.8±1.5 |
| CRow | 54.5±3.7 | 42.6±3.2 | 52.5±4.5 | 38.0±3.7 | 39.6±1.5 | 42.3±3.5 | 72.1±2.7 |
| TimeGAN | 50.0±1.8 | 44.5±19.3 | 48.8±1.3 | 37.6±1.8 | 40.3±2.5 | 43.4±0.8 | 72.4±1.3 |
| CTAB-GAN | 59.5±6.7 | 48.6±4.8 | 59.1±5.5 | 45.0±2.3 | 49.9±7.3 | 48.0±3.2 | 77.0±3.3 |
| CTAB-GAN+ | 52.2±2.6 | 45.8±0.0 | 53.3±2.2 | 38.8±1.7 | 52.3±4.4 | 43.3±0.7 | 73.3±0.4 |
| DDPM | 55.7±2.6 | 45.8±14.4 | 54.7±1.3 | 41.3±2.9 | 41.4±2.7 | 43.8±1.1 | 75.4±1.1 |
| ConDOR | 57.0±3.5 | 50.0±7.2 | 55.4±5.7 | 42.3±1.4 | 51.6±19.0 | 47.0±1.2 | 75.0±3.2 |
| NOHA (Ours) | 64.9±1.9 | 51.4±4.8 | 61.6±4.5 | 49.0±1.6 | 55.4±13.4 | 51.4±0.2 | 78.3±3.1 |

*Table 2.* Classification performance using a Transformer. The average of three replicates and their standard deviations are reported. The best results are in bold, and the second-best are underlined.

| Methods | Longitudinal evaluation | | Cross-sectional evaluation | | | | |
|---|---|---|---|---|---|---|---|
| | Accuracy | | Accuracy | F1 | Precision | Recall | Specificity |
| | All | Transition-only | | | | | |
| **Amyloid** | | | | | | | |
| No Aug. | 53.8±1.0 | 54.9±2.3 | 53.0±2.7 | 50.0±3.0 | 60.0±2.9 | 49.0±2.5 | 72.0±0.2 |
| SMOTE | 45.4±5.8 | 36.8±3.5 | 46.0±5.5 | 31.6±4.4 | 53.8±4.2 | 38.9±3.6 | 69.1±1.1 |
| TVAE | 49.0±3.1 | 41.1±8.1 | 49.7±1.3 | 36.2±1.6 | 45.2±15.2 | 41.3±3.4 | 69.5±0.5 |
| GOGGLE | 45.4±8.6 | 36.8±21.5 | 46.4±6.8 | 32.2±9.7 | 44.6±16.9 | 41.8±8.5 | 69.0±2.6 |
| CRow | 39.6±6.7 | 33.3±13.3 | 38.9±9.5 | 41.3±29.5 | 40.9±6.9 | 28.1±7.1 | 69.3±2.3 |
| TimeGAN | 59.0±2.1 | 54.3±10.3 | 57.0±1.7 | 48.7±6.3 | 72.8±13.8 | 50.0±5.7 | 74.4±2.2 |
| CTAB-GAN | 49.9±5.0 | 43.7±4.3 | 50.6±6.5 | 37.8±2.2 | 46.9±11.2 | 44.7±4.4 | 71.3±2.5 |
| CTAB-GAN+ | 45.9±5.7 | 44.7±4.8 | 45.5±7.2 | 31.7±5.3 | 41.9±14.4 | 42.2±6.3 | 69.2±1.3 |
| DDPM | 57.3±3.0 | 55.5±10.2 | 56.4±2.3 | 49.2±8.0 | 62.1±5.9 | 51.0±7.9 | 73.7±2.8 |
| ConDOR | 56.2±3.2 | 57.2±6.3 | 56.0±2.6 | 45.8±6.2 | 73.8±3.5 | 49.5±4.5 | 72.4±2.0 |
| NOHA (Ours) | 60.0±1.0 | 64.0±1.7 | 60.5±1.1 | 57.5±2.3 | 67.1±2.0 | 55.4±2.1 | 75.9±1.2 |
| **Tau (ADNI + OASIS)** | | | | | | | |
| No Aug. | 62.7±3.0 | 54.2±2.0 | 61.6±3.1 | 47.1±0.5 | 47.6±2.7 | 49.6±1.6 | 78.3±1.7 |
| SMOTE | 62.0±6.2 | 51.4±8.7 | 62.7±6.6 | 45.6±4.5 | 51.4±11.1 | 49.6±4.7 | 78.3±3.9 |
| TVAE | 49.0±2.9 | 41.1±8.3 | 49.7±1.3 | 36.2±1.6 | 45.2±15.2 | 41.3±3.4 | 69.5±0.5 |
| GOGGLE | 45.4±8.4 | 36.8±21.4 | 46.4±6.8 | 32.2±9.7 | 44.6±16.9 | 41.8±8.5 | 69.0±2.6 |
| CRow | 39.6±6.4 | 33.3±13.3 | 38.9±9.5 | 41.3±29.5 | 40.9±6.9 | 28.1±6.9 | 69.3±2.0 |
| TimeGAN | 51.6±5.5 | 38.9±9.6 | 49.2±6.9 | 39.3±2.7 | 41.6±5.3 | 42.7±0.8 | 71.6±1.9 |
| CTAB-GAN | 49.9±5.0 | 43.7±4.3 | 51.0±6.5 | 37.8±2.2 | 46.9±11.1 | 44.7±4.4 | 71.3±2.5 |
| CTAB-GAN+ | 45.9±5.7 | 44.7±4.8 | 45.5±7.2 | 32.0±5.3 | 41.9±14.5 | 42.2±6.4 | 69.0±1.3 |
| DDPM | 58.5±3.7 | 47.2±6.4 | 58.3±2.5 | 44.9±2.5 | 52.3±3.4 | 47.3±2.3 | 76.5±1.9 |
| ConDOR | 63.0±4.1 | 48.6±4.8 | 61.2±6.0 | 48.2±2.6 | 51.9±7.3 | 50.4±2.2 | 78.5±3.6 |
| NOHA (Ours) | 63.5±4.8 | 55.6±2.4 | 63.4±6.0 | 52.5±0.4 | 56.6±7.2 | 53.4±1.6 | 79.5±4.0 |

*Table 3.* Time for generating $N_{aug} = 210$ sequences in the Amyloid experiment.

| Model | Time (sec) ↓ |
|---|---|
| SMOTE | **0.151** |
| TVAE | 6.791 |
| GOGGLE | 8.968 |
| CRow | 3.197 |
| TimeGAN | 2.939 |
| CTAB-GAN | 92.548 |
| CTAB-GAN+ | 5.092 |
| DDPM | 87.144 |
| ConDOR | 36.029 |
| NOHA | 0.491 |

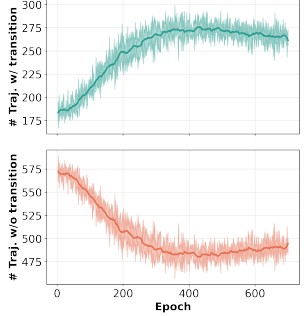

*Figure 2.* Number of transition and non-transition FDG samples over training epochs.

faster than CTAB-GAN. These results validate the efficiency of NOHA, which enables scalable longitudinal data generation with minimal computational overhead.

### 4.4. Model Behavior Analysis

#### 4.4.1. EFFECT OF HAZARD-GUIDED SAMPLING

Figure 2 illustrates the changes in the number of transition and non-transition samples, including both the original and generated samples, throughout training on the FDG experiment. As training progresses, our method increasingly prioritizes time windows with high transition likelihood, leading to a gradual rise in the number of transition samples and a corresponding decrease in non-transition samples. This behavior indicates that the hazard-guided sampling effectively adjusts the sampling distribution according to the learned transition dynamics to emphasize rare but informative disease progression events. By progressively focusing on these label-changing intervals, the model alleviates trajectory-level data imbalance (i.e., transition vs. non-transition) and

learns richer disease progression patterns during training.

The quantitative impact of this hazard-guided sampling is summarized in Table 4. By randomly sampling the subject-interval pair $(i, k)$, our method showed improvement over the non-augmentation setting ('No Aug') in most metrics; however, hazard-guided sampling consistently surpasses the random sampling across all metrics and biomarkers.

In particular, it yields notable gains in transition-only accuracy, with a 20.4%p accuracy improvement in the cortical thickness experiment. This validates that explicitly guiding the sampling process toward transition-prone intervals substantially enhances the model's ability to capture temporal dynamics associated with disease progression. Moreover, consistent improvements in cross-sectional evaluations indicate enhanced discriminability of time-point–level feature representations. Overall, this ablation study demonstrates that the performance gains of NOHA are not merely attributed to increased data volume but stem from the proposed hazard-guided sampling mechanism that selectively

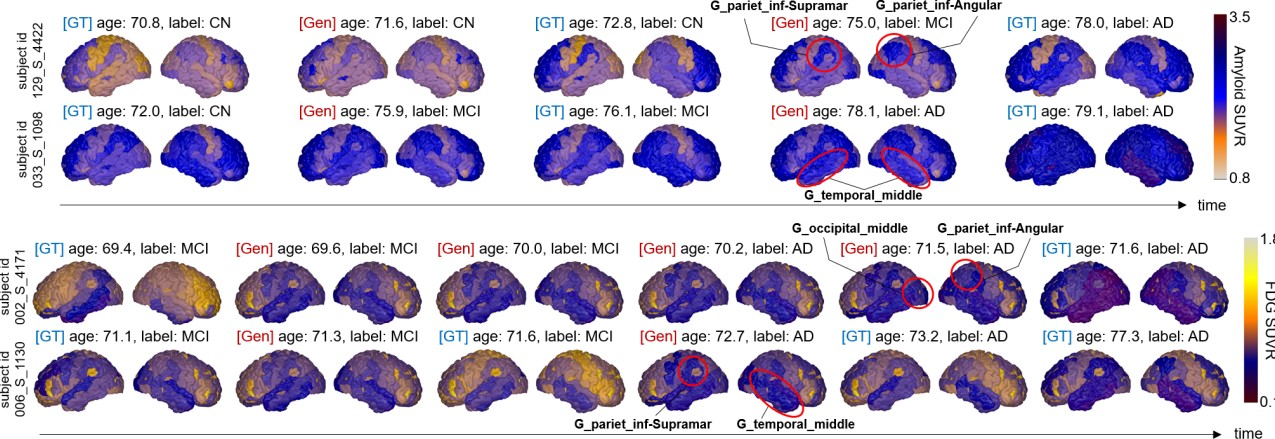

*Figure 3.* Visualization of ground-truth and generated longitudinal (Top) amyloid and (Bottom) FDG SUVR trajectories. For each subject, brain maps are ordered by age, showing both ground-truth (GT, blue) and generated (Gen, red) samples. NOHA produces age- and label-consistent trajectories with realistic amyloid accumulation and FDG reduction patterns in AD-related brain regions (marked in red circles) and clinically plausible diagnostic transitions from CN, MCI, to AD.

*Table 4.* Ablation study on the hazard-guided sampling using an RNN classifier.

| Methods | Longitudinal eval. Accuracy | | Cross-sectional eval. | | | | |
|---|---|---|---|---|---|---|---|
| | All | Trans.-only | Acc. | F1 | Prec. | Recall | Spec. |
| **Cortical Thickness** | | | | | | | |
| No Aug. | $65.9^{\pm 9.6}$ | $50.0^{\pm 28.9}$ | $64.1^{\pm 7.9}$ | $58.0^{\pm 3.0}$ | $65.5^{\pm 1.0}$ | $62.9^{\pm 6.2}$ | $79.3^{\pm 3.8}$ |
| Ours w/ random sampling | $63.8^{\pm 7.9}$ | $46.3^{\pm 11.6}$ | $63.4^{\pm 7.4}$ | $59.3^{\pm 2.4}$ | $70.6^{\pm 6.2}$ | $62.1^{\pm 7.0}$ | $78.3^{\pm 3.6}$ |
| Ours w/ hazard-guided sampling | $73.0^{\pm 2.8}$ (+ 9.2) | $66.7^{\pm 0.0}$ (+ 20.4) | $71.9^{\pm 2.3}$ (+ 8.5) | $73.1^{\pm 1.0}$ (+ 13.8) | $82.7^{\pm 6.5}$ (+ 12.1) | $75.0^{\pm 7.0}$ (+ 12.9) | $83.8^{\pm 1.9}$ (+ 5.5) |
| **Amyloid** | | | | | | | |
| No Aug. | $57.3^{\pm 2.8}$ | $52.0^{\pm 6.3}$ | $56.4^{\pm 2.5}$ | $49.8^{\pm 7.1}$ | $65.4^{\pm 4.0}$ | $49.1^{\pm 5.8}$ | $73.1^{\pm 2.0}$ |
| Ours w/ random sampling | $57.7^{\pm 1.2}$ | $62.1^{\pm 8.0}$ | $57.2^{\pm 2.1}$ | $52.6^{\pm 1.0}$ | $66.2^{\pm 9.4}$ | $52.5^{\pm 1.9}$ | $73.7^{\pm 0.5}$ |
| Ours w/ hazard-guided sampling | $61.9^{\pm 0.9}$ (+ 4.2) | $69.2^{\pm 3.3}$ (+ 7.1) | $61.8^{\pm 0.9}$ (+ 4.6) | $57.1^{\pm 1.1}$ (+ 4.5) | $68.9^{\pm 5.1}$ (+ 2.6) | $57.1^{\pm 2.4}$ (+ 4.6) | $76.7^{\pm 1.0}$ (+ 3.0) |
| **FDG** | | | | | | | |
| No Aug. | $50.7^{\pm 2.0}$ | $42.4^{\pm 10.3}$ | $52.7^{\pm 1.4}$ | $46.6^{\pm 7.0}$ | $62.5^{\pm 5.5}$ | $50.0^{\pm 7.0}$ | $74.5^{\pm 2.9}$ |
| Ours w/ random sampling | $52.7^{\pm 2.3}$ | $50.3^{\pm 2.1}$ | $52.7^{\pm 3.0}$ | $47.6^{\pm 9.7}$ | $55.9^{\pm 3.5}$ | $48.6^{\pm 7.6}$ | $73.5^{\pm 3.4}$ |
| Ours w/ hazard-guided sampling | $61.5^{\pm 1.6}$ (+ 8.8) | $55.4^{\pm 2.2}$ (+ 5.1) | $61.5^{\pm 2.4}$ (+ 8.8) | $60.9^{\pm 3.0}$ (+ 13.3) | $63.2^{\pm 3.0}$ (+ 7.3) | $60.5^{\pm 3.3}$ (+ 11.9) | $79.0^{\pm 1.0}$ (+ 5.5) |
| **Tau (ADNI + OASIS)** | | | | | | | |
| No Aug. | $58.9^{\pm 3.5}$ | $34.7^{\pm 9.6}$ | $57.6^{\pm 1.9}$ | $40.1^{\pm 2.8}$ | $45.1^{\pm 1.9}$ | $45.4^{\pm 2.1}$ | $75.8^{\pm 1.8}$ |
| Ours w/ random sampling | $60.5^{\pm 4.6}$ | $50.0^{\pm 7.2}$ | $59.1^{\pm 4.9}$ | $44.2^{\pm 4.6}$ | $43.1^{\pm 4.6}$ | $47.2^{\pm 3.7}$ | $77.3^{\pm 2.8}$ |
| Ours w/ hazard-guided sampling | $64.9^{\pm 1.9}$ (+ 4.4) | $51.4^{\pm 4.8}$ (+ 1.4) | $61.6^{\pm 4.5}$ (+ 2.5) | $49.0^{\pm 1.6}$ (+ 4.8) | $55.4^{\pm 13.4}$ (+ 12.3) | $51.4^{\pm 0.2}$ (+ 4.2) | $78.3^{\pm 3.1}$ (+ 1.0) |

focuses on clinically critical transition events.

### 4.4.2. QUALITATIVE EVALUATION OF SYNTHESIZED TRAJECTORY AND BRAIN REGIONAL ANALYSIS

To assess the biological validity of the synthesized data by NOHA, we qualitatively compare ground-truth and generated longitudinal trajectories for amyloid and FDG SUVR in Figure 3. Across subjects, the generated trajectories remain consistent with age and diagnostic labels, and exhibit plausible AD progression patterns, including increasing amyloid deposition and decreasing FDG uptake over time.

Specifically, NOHA accurately captures pronounced Amyloid accumulation and FDG hypometabolism patterns in the inferior parietal lobule, including the *supramarginal gyrus* and the *angular gyrus*, as well as the *middle temporal gyrus*. These ROIs have been widely reported to show early amyloid accumulation in AD patients (Collij et al., 2022), and exhibit strong spatial correspondence with FDG

hypometabolism (Kato et al., 2016), supporting the close coupling between amyloid pathology and glucose metabolic decline in AD in the parietal lobe (Ossenkoppele et al., 2012). Furthermore, NOHA successfully identifies a reduction in FDG SUVR within the *middle occipital gyrus*. Such metabolic decline is consistent with prior studies (Sawyer & Kuo, 2018; Rehak et al., 2016), in which hypometabolism predominantly affects occipital and temporoparietal regions.

Notably, in both Amyloid and FDG trajectories, such key regional patterns do not clearly stand out in the earlier disease stages, but NOHA successfully synthesized the intermediate missing links by assuming the underlying individual pathological evolution at unobserved time points. These results highlight the potential of NOHA to simulate the continuous spectrum of disease progression beyond the discrete and sparse observations available in the real-world clinical data.

## 5. Conclusion

We proposed NOHA, a novel generative framework that integrates a CTMC with Neural ODE to model non-linear and irreversible longitudinal dynamics with rare label transitions. In this work, NOHA is applied to characterize degenerative disease progression under sparse and heterogeneous observations across individuals. By employing a hazard-guided sampling, NOHA explicitly accounts for time-varying state-transition risks, thereby prioritizing critical yet infrequent disease state transitions. Moreover, NOHA jointly synthesizes biomarker trajectories, patient attributes, and diagnostic labels, ensuring temporal consistency across multiple time-varying factors. Extensive evaluations on four biomarkers validate that NOHA synthesizes high-fidelity trajectories with realistic pathological progression patterns, highlighting its potential to enrich the understanding of disease transition dynamics under sparse and irregular observations.

## Impact Statement

This paper presents work whose goal is to advance the field of machine learning and its application to longitudinal medical data analysis to understand the progress of degenerative diseases. There are many potential societal consequences of our work, especially given the scope of our application, such as simulating irreversible disease progression within longitudinal patient trajectories, alleviating data scarcity in medical research, and facilitating the robust development of downstream models for disease progression forecasting.

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

In the appendix, we present **1)** pseudo-code of NOHA, **2)** dataset demographics of ADNI and OASIS, **3)** detailed implementation setup for all experiments, **4)** additional qualitative results, and **5)** limitation and future work, which were not included in the main manuscript due to the page limit.

## A. Pseudo-code

---

**Algorithm 1** Training NOHA with Hazard-Guided Trajectory Sampling

---

**Input:** Training dataset $\mathcal{D}$ including Longitudinal data $x_i$, Labels $y_i$, Covariates $\Gamma_i = \gamma_i \cup \bar{\gamma}_i$, where Age $a_i \in \gamma_i$.
**Output:** Augmented dataset $\mathcal{D}_{aug}$ including Synthesized longitudinal data $\tilde{x}_i$, Pseudo-labels $\tilde{y}_i$, Covariates $\tilde{\Gamma}_i = \tilde{\gamma}_i \cup \bar{\gamma}_i$.

```
// Step 1:  Longitudinal Data Encoding and ODE Reconstruction
```
$z_{i,1} \leftarrow \text{Enc}(x_{i,1}, y_{i,1}, \gamma_{i,t}, \bar{\gamma}_i)$      `// Encode initial latent states`
$z_{i,k} \leftarrow \text{ODESolve}(z_{i,1}, f_\theta, a_{i,1}, t)$      `// Solve Neural ODE`
$\hat{x}_{i,k} \leftarrow \text{Decoder}(z_{i,k}, y_{i,k}, \gamma_{i,t}, \bar{\gamma}_i)$      `// Data Reconstruction`
$L_{\text{rec}} = \mathbb{E}_{i,k}\big[\|\hat{x}_{i,k} - x_{i,k}\|_2^2\big]$

```
// Step 2:  CTMC-based Transition Hazard Estimation
```
$\{q_{i,k}^{c \to c+1}\}_{c=1}^{C-1} \leftarrow q_\phi(z_{i,k}, y_{i,k}, a_{i,k}, a_{i,k+1})$      `// Estimate temporal transition hazards`
$p_{i,k}^{c \to c+1} \leftarrow 1 - \exp(-q_{i,k}^{c \to c+1} \cdot \Delta a_{i,k}) \in [0,1]$      `// Compute transition probabilities`
$y_{i,k}' = \mathbf{1}[y_{i,k} = c \wedge y_{i,k+1} = c+1]$
$m_{i,k} = \mathbf{1}[y_{i,k} \in \{1, \ldots C-1\}] \cdot \mathbf{1}[k < K_i]$
$L_{\text{hazard}} = \mathbb{E}_{i,k}\big[m_{i,k} \cdot -\big(y_{i,k}' \log p_{i,k}^{c \to c+1} + (1 - y_{i,k}') \log(1 - p_{i,k}^{c \to c+1})\big)\big]$

```
// Step 3:  Hazard-Guided Trajectory Sampling
```
$N_{\text{aug}} \leftarrow N_{\text{trans}} - N_{\text{w/o trans}}$
**for** $n = 1$ **to** $N_{aug}$ **do**
     Sample $(i,k) \sim \Pr(i,k) = \frac{m_{i,k} \cdot p_{i,k}^{c \to c+1}}{\sum_{i'} \sum_{k'} m_{i',k'} \cdot p_{i',k'}^{c \to c+1}}$      `// Sampling subject-interval pairs`
     **for** *each sampled pair* $(i,k)$ **do**
$$\tilde{a}_{i,1:\tilde{K}_i} = \begin{cases} \tilde{a}_{i,1} = a_{i,\text{start}} \in \{a_{i,j}\}_{j=0}^{k}, \\ \tilde{a}_{i,\tilde{K}_i} \in \{a_{i,j}\}_{j=k+1}^{K_i}, \\ \tilde{a}_{i,2:\tilde{K}_i-1} \sim \text{Uniform}(\tilde{a}_{i,1}, \tilde{a}_{i,\tilde{K}_i}) \end{cases}$$      `// Pseudo-time point sampling`
         $\{\tilde{q}_{i,\tilde{k}}^{c \to c+1}\}_{c=1}^{C-1} \leftarrow g_\phi([\tilde{z}_{i,\tilde{k}}, \tilde{y}_{i,\tilde{k}}, \tilde{a}_{i,\tilde{k}}, \tilde{a}_{i,\tilde{k}+1}])$      `// Estimate temporal transition hazards`
         $\tilde{p}_{i,\tilde{k}}^{c \to c+1}(\Delta \tilde{a}_{i,\tilde{k}}) \leftarrow 1 - \exp(-\tilde{q}_{i,\tilde{k}}^{c \to c+1} \Delta \tilde{a}_{i,\tilde{k}})$      `// Compute transition probabilities`
         $\tilde{y}_{i,\tilde{1}} = y_{i,\text{start}}$
$$\tilde{y}_{i,\tilde{k}+1} = \begin{cases} \tilde{y}_{i,\tilde{k}}, & \text{if } (u \geq \tilde{p}_{i,\tilde{k}}^{c \to c+1}) \wedge (\tilde{y}_{i,\tilde{k}} < C), \\ \tilde{y}_{i,\tilde{k}} + 1, & \text{if } (u < \tilde{p}_{i,\tilde{k}}^{c \to c+1}) \wedge (\tilde{y}_{i,\tilde{k}} < C), \\ C, & \text{if } (\tilde{y}_{i,\tilde{k}} = C), \end{cases}$$      `// Pseudo-label sampling`
         $\tilde{z}_{i,1} = z_{i,\text{start}}$
         $\tilde{z}_{i,k} \leftarrow \text{ODESolve}(\tilde{z}_{i,1}, f_\theta, \tilde{a}_{i,1}, t)$      `// Solve Neural ODE`
         $\tilde{x}_{i,\tilde{k}} \leftarrow \text{Dec}(\tilde{z}_{i,\tilde{k}}, \tilde{y}_{i,\tilde{k}}, \tilde{\gamma}_{i,\tilde{k}}, \bar{\gamma}_i)$      `// Data Reconstruction`
     **end**
**end**

```
// Step 4:  Joint Optimization
```
$\mathcal{D}_{\text{total}} \leftarrow \text{concat}(\mathcal{D}, \mathcal{D}_{\text{aug}})$
$L_{\text{cls}} \leftarrow \text{Classifier}(\mathcal{D}_{\text{total}})$      `// Autoregressive sequence classification`
$\mathcal{L} \leftarrow L_{\text{recon}} + L_{\text{hazard}} + L_{\text{cls}}$      `// Update parameters via gradient descent`

---

## B. Dataset Demographics

### B.1. ADNI Dataset

ADNI (Mueller et al., 2005) offers extensive multisite longitudinal neuroimaging and genetic data. In this study, we focused on four key AD-related biomarkers: (1) cortical thickness (CT) derived from MRI, and standardized uptake value ratios (SUVR) of (2) Amyloid, (3) fluorodeoxyglucose (FDG), and (4) Tau derived from PET scans. For neuroimaging data, we utilized the Destrieux atlas (Destrieux et al., 2010) to parcellate the brain into 148 regions. T1-weighted MR images underwent skull stripping, tissue segmentation, and registration to measure region-wise average cortical thickness using FreeSurfer. For PET modalities (Amyloid, FDG, and Tau), we computed the regional average SUVR using the cerebellum as the reference region.

The ADNI database supplies SNP genotype data in PLINK format, assayed via the Illumina method, consisting of 620,901 SNPs for the ADNI-1 phase and 730,525 SNPs for the ADNIGO/2 phases. We processed Single Nucleotide Polymorphism (SNP) data provided in PLINK format. To ensure data quality, we performed rigorous quality control (QC) filtering: samples were excluded if they had a missing genotype rate $> 5\%$, a minor allele frequency $< 1\%$, or failed the Hardy-Weinberg exact test ($p < 10^{-5}$). From the retained SNPs, we specifically selected 49 markers linked to AD candidates listed in the AlzGene database (Bertram et al., 2007). Additionally, since the APOE $\epsilon4$ allele is a critical risk factor, we extracted two APOE $\epsilon4$-relevant SNPs (rs429368 and rs7412). The final 51 genotype features were encoded into one of three categories: 0 (normal), 1 (heterogeneous variant), or 2 (homogeneous variant).

We curated our dataset by selecting subjects who had both longitudinal imaging data with at least two time points and corresponding covariates. For CT, Amyloid, and FDG, we utilized age, gender, and genotype as covariates, where age is a temporal covariate that corresponds to the time point at which the biomarker was obtained. For the Tau modality, we incorporated domain information as a conditioning variable instead of genotype data, since we conducted an independent multi-domain experiment by integrating the ADNI and OASIS cohorts, and genotype information is not available for OASIS subjects. After filtering out subjects with only one time point, the final dataset comprised 102 subjects for CT, 478 for Amyloid, 642 for FDG, and 160 for Tau. The detailed demographic statistics are presented in Table 5. The maximum number of time points was five for CT and Tau, six for Amyloid, and ten for FDG. The average time intervals between data acquisitions were calculated as follows: $2.27 \pm 0.91$ years for CT, $2.19 \pm 0.78$ years for Amyloid, $1.17 \pm 1.05$ years for FDG, and $1.19 \pm 0.38$ years for Tau.

### B.2. OASIS Dataset

We also employed the OASIS-3 dataset (LaMontagne et al., 2019), which includes Tau-PET scans. Similar to the ADNI preprocessing pipeline, we registered PET scans to T1-weighted images and calculated regional SUVRs based on the Destrieux atlas, using the cerebellum cortex as a reference. We selected 32 subjects who possessed complete demographic

*Table 5.* Demographics of the ADNI dataset.

| Biomarker | Category | Longitudinal data demographics | | Cross-sectional data demographics | | |
| --- | --- | --- | --- | --- | --- | --- |
| | | Traj. w/ Trans. | Traj. w/o Trans. | CN | MCI | AD |
| Cortical Thickness | # of subjects | 16 | 86 | 51 | 47 | 4 |
| | Gender (M / F) | 9/7 | 55/31 | 30/21 | 30/17 | 4/0 |
| | Age (Mean $\pm$ Std) | $73.5 \pm 7.7$ | $71.7 \pm 6.4$ | $73.8 \pm 5.4$ | $69.4 \pm 6.6$ | $80.2 \pm 7.6$ |
| Amyloid | # of subjects | 104 | 374 | 185 | 251 | 42 |
| | Gender (M / F) | 60/44 | 219/155 | 104/81 | 149/102 | 26/16 |
| | Age (Mean $\pm$ Std) | $74.0 \pm 6.6$ | $72.5 \pm 7.0$ | $74.1 \pm 5.6$ | $71.6 \pm 7.5$ | $74.7 \pm 7.6$ |
| FDG | # of subjects | 156 | 486 | 222 | 323 | 97 |
| | Gender (M / F) | 99/57 | 301/185 | 130/92 | 212/111 | 58/39 |
| | Age (Mean $\pm$ Std) | $74.6 \pm 6.7$ | $74.1 \pm 6.9$ | $75.2 \pm 5.3$ | $73.1 \pm 7.5$ | $75.9 \pm 7.1$ |
| Tau | # of subjects | 6 | 154 | 60 | 80 | 20 |
| | Gender (M / F) | 2/4 | 86/68 | 32/28 | 45/35 | 11/9 |
| | Age (Mean $\pm$ Std) | $75.8 \pm 3.9$ | $70.8 \pm 6.9$ | $72.1 \pm 5.9$ | $69.9 \pm 7.1$ | $71.6 \pm 8.5$ |

*Table 6.* Demographics of the OASIS dataset.

| Biomarker | Category | Longitudinal data demographics | | Cross-sectional data demographics | |
|---|---|---|---|---|---|
| | | Traj. w/ Trans. | Traj. w/o Trans. | CN | AD |
| Tau | # of subjects | 2 | 30 | 31 | 1 |
| | Gender (M / F) | 1/1 | 13/17 | 13/18 | 1/0 |
| | Age (Mean $\pm$ Std) | $70.0 \pm 0.0$ | $63.1 \pm 7.7$ | $63.2 \pm 7.5$ | $73.0 \pm$ n/a |

labels and longitudinal scans. All subjects in this cohort had exactly two time points separated by a 6-month interval. Among them, three subjects were diagnosed with AD, including one stable AD and two converters from CN. The demographics for the OASIS cohort are summarized in Table 6.

## C. Detailed Implementation Setup

To train NOHA, we employed Pytorch with a single NVIDIA GeForce RTX 3090. In Table 7, we provide details of the implementation settings of NOHA for all experiments on four biomarkers. We performed a grid search to choose the best learning rate in {0.1, 0.01}, batch size in {4, 8, 16, 32}, hidden dimension and the final latent feature dimension in {8, 16, 32, 64}. The source code of NOHA will be released online.

For all biomarkers, we applied a stratified train-test split to the data, maintaining the ratio of the given three labels, with 80% of data used for training and the remaining 20% for testing. Also, for the multi-domain learning on Tau, we applied a stratified train-test split to maintain the sample size ratio of each domain. To prevent biased results, all experiments were replicated three times with different parameter initialization, and their averaged results along with standard deviations were reported.

*Table 7.* Hyperparameters of NOHA for all experiment settings.

| | Hyperparameter | CT | Amyloid | FDG | Tau |
|---|---|---|---|---|---|
| Train | Optimizer | SGD | SGD | SGD | SGD |
| | Learning rate | 0.1 | 0.1 | 0.1 | 0.1 |
| | Learning rate scheduler | LambdaLR | LambdaLR | LambdaLR | LambdaLR |
| | Learning rate decay | 0.995 | 0.995 | 0.995 | 0.995 |
| | Batch size | 16 | 32 | 8 | 4 |
| | Number of epochs | 800 | 800 | 800 | 800 |
| Enc($\cdot$) | Number of layers | 3 | 3 | 3 | 3 |
| | Hidden dimensions | [8, 4] | [64, 32] | [16, 8] | [32, 16] |
| | Final latent feature dimension | 16 | 32 | 64 | 8 |
| | activation function | ReLU | ReLU | ReLU | ReLU |
| Dec($\cdot$) | Number of layers | 3 | 3 | 3 | 3 |
| | Hidden dimensions | [2, 4] | [16, 32] | [4, 8] | [8, 16] |
| | activation function | ReLU | ReLU | ReLU | ReLU |
| | Initial latent feature dimension | 16 | 32 | 64 | 8 |
| $f_\theta$ | Number of layers | 2 | 2 | 2 | 2 |
| | Hidden dimension | 8 | 64 | 16 | 32 |
| | activation function | Tanh | Tanh | Tanh | Tanh |
| | Initial latent feature dimension | 16 | 32 | 64 | 8 |
| | Final latent feature dimension | 16 | 32 | 64 | 8 |
| | Solver | Heun | Heun | Heun | Heun |
| $g_\phi$ | Number of layers | 2 | 2 | 2 | 2 |
| | Hidden dimension | 4 | 32 | 8 | 16 |
| | activation function | ReLU | ReLU | ReLU | ReLU |
| | Initial latent feature dimension | 19 (16+3) | 35 (32+3) | 67 (64+3) | 11 (8+3) |
| | Final latent feature dimension | 1 | 1 | 1 | 1 |

## D. Additional Qualitative Results

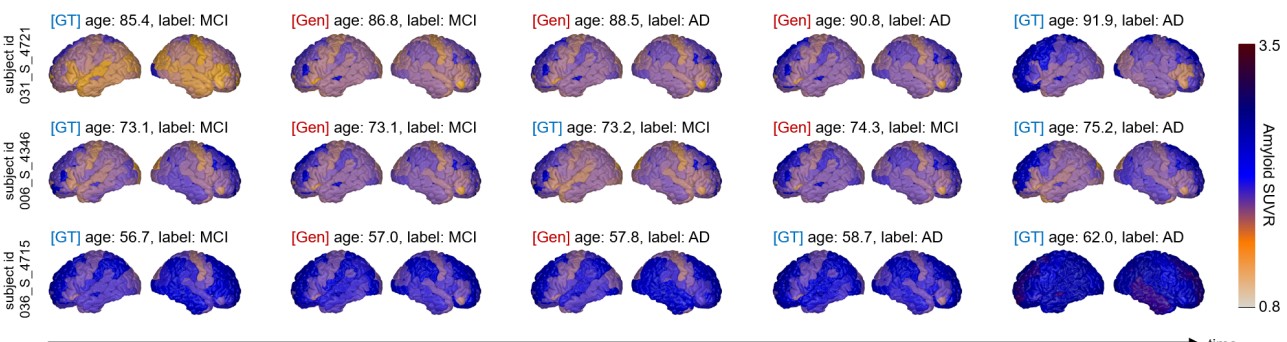

*Figure 4.* Visualization of ground-truth and generated longitudinal amyloid SUVR trajectories. For each subject, brain maps are ordered by age, showing both ground-truth (GT, blue) and generated (Gen, red) samples. NOHA produces age- and label-consistent trajectories with realistic amyloid accumulation and clinically plausible diagnostic transitions from CN, MCI, to AD.

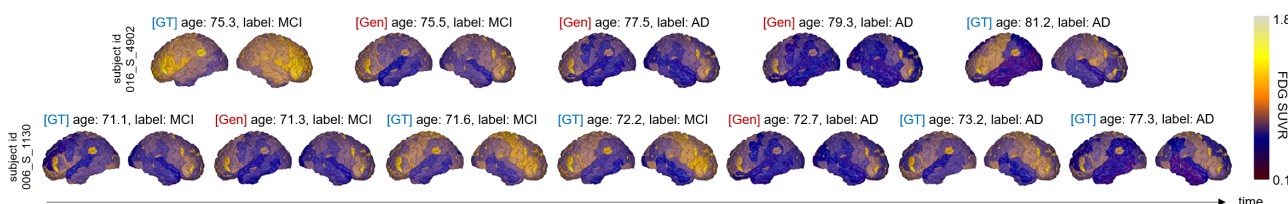

*Figure 5.* Visualization of ground-truth and generated longitudinal FDG SUVR trajectories. For each subject, brain maps are ordered by age, showing both ground-truth (GT, blue) and generated (Gen, red) samples. NOHA produces age- and label-consistent trajectories with realistic FDG reduction patterns and clinically plausible diagnostic transitions from CN, MCI, to AD.

## E. Limitation and Future Work

Despite the promising performance of NOHA in synthesizing longitudinal biomarker trajectories, several limitations and future directions remain to be addressed.

**(1) Limited scope of evaluation domain.** In this work, the proposed method is evaluated only on Alzheimer's disease (AD) datasets, which may limit the generalizability of NOHA. While the effectiveness of NOHA is demonstrated on four key AD-related biomarkers, the applicability of NOHA to other neurodegenerative disorders, such as Parkinson's disease, remains to be investigated. Moreover, NOHA is not restricted to medical applications and can potentially be extended to non-medical domains where longitudinal data exhibit rare label transitions. For instance, in financial risk modeling or industrial predictive maintenance, critical state-transition events occur infrequently, such as market crashes or sudden mechanical failures. If these state-transitions are irreversible, evaluating NOHA in these non-medical contexts would provide a more rigorous assessment of the generality and robustness of its hazard-guided sampling strategy.

**(2) Lack of high-dimensional data modeling.** NOHA focuses on trajectory-level synthesis of region-wise one-dimensional biomarker measurements rather than image-level data, and therefore does not directly model spatial correlations at the voxel or image level. Extending the proposed framework to 2D image-level, or even 3D longitudinal synthesis remains a promising direction for future work.

