# OpenReview forum: "Hazard-Guided Generative Modeling for Sparse and Irreversible Transitions in Longitudinal Disease Trajectories"
_ICML.cc/2026/Conference — Submitted to ICML 2026_

### Official Review · Reviewer_5AHh · 2026-02-17

**Soundness:** 3
**Presentation:** 4
**Significance:** 4
**Originality:** 3
**Overall Recommendation:** 4
**Confidence:** 4

**Summary:**

This paper proposes the NOHA framework to address the challenge of modeling sparse and irreversible disease progression events in longitudinal medical data. By combining Neural ODE and continuous-time Markov chains, this model captures nonlinear and continuous disease dynamics and employs a hazard-guided sampling strategy to prioritize the synthesis of high-risk disease state transition trajectories. NOHA not only handles irregular sampling but also simultaneously generates evolving patient attributes and diagnostic labels, significantly improving the clinical fidelity of the synthesized data and its utility in downstream prediction tasks.

The authors' core contributions and innovations are as follows:
* Addressing the class imbalance problem in clinical data caused by the significantly lower number of "transition" events compared to "stable" states, NOHA introduces the concept of "hazard" from survival analysis. By estimating the immediate transition risk of subjects within specific time intervals, the model can proactively "sample" and prioritize generating clinically significant but infrequent disease state transition events, rather than treating all time points uniformly.
* To handle irregular time intervals in longitudinal data, NOHA utilizes Neural ODE to model underlying nonlinear, continuous-time disease dynamics. Simultaneously, it combines CTMC to capture the irreversible transition risk between discrete disease states (such as from healthy to mild cognitive impairment). This design allows the model to simulate both continuous biomarker changes and discrete state transitions.
* Unlike existing models that typically treat patient attributes (such as age) and diagnostic labels as static inputs, NOHA achieves the joint generation of biomarker trajectories, time-evolving covariates, and disease labels. This mechanism ensures high clinical consistency of the synthetic data over time.
* The authors tested four key biomarkers on two neurodegenerative disease datasets (such as Alzheimer's disease). The results show that the high-fidelity trajectories generated by NOHA not only capture rare progression events but also significantly improve the performance of downstream disease progression prediction tasks, providing an effective tool for addressing the problems of scarce clinical data and class imbalance.

**Compliance With Llm Reviewing Policy:**

Affirmed.

**Final Justification:**

Following a thorough review of the manuscript and the authors' rebuttal, I maintain my recommendation of Weak Accept (4). The paper presents a technically sound and clinically relevant framework, NOHA, which effectively addresses the critical challenge of modeling sparse and irreversible disease progression in longitudinal datasets.

**Key Questions For Authors:**

* Q1: Diagnostic labels in neurodegenerative diseases are often "noisy" or subject to inter-rater variability. Since NOHA uses these labels to estimate transition hazards, how does the model handle label noise or misclassification in the training set? Would a few mislabeled stable patients significantly bias the hazard-guided sampling distribution?
* Q2: The model incorporates covariates and domain labels as conditioning variables. While this is a standard architecture, it is unclear if this is sufficient to mitigate confounding effects or domain shifts. Have the authors performed any quantitative analysis to verify that the latent space is truly invariant to these effects while remaining sensitive to biological progression?

**Limitations:**

yes

**Strengths And Weaknesses:**

Strengths:
* S1: By integrating the instantaneous transition hazard from a Continuous-Time Markov Chain (CTMC) with the continuous latent dynamics of Neural ODEs, NOHA provides a rigorous mathematical framework for modeling irreversible disease progression.
* S2: The proposed Hazard-Guided Sampling strategy adaptively prioritizes and augments underrepresented state-transition samples. This is crucial in clinical practice for capturing critical, yet sparse, windows of disease aggravation.
* S3: Unlike prior generative models that treat covariates (e.g., age) as static conditions, NOHA jointly generates time-varying covariates, labels, and biomarkers. This ensures that the synthesized trajectories maintain high clinical consistency across multiple evolving factors.
* S4: The framework was rigorously benchmarked against nine mainstream generative models across four distinct biomarkers, demonstrating significant performance gains in downstream longitudinal classification tasks.

Weaknesses:
* W1: The framework essentially functions as a hazard-guided data augmentation scheme. However, the evaluation of the synthesized longitudinal trajectories relies almost exclusively on a single downstream task: disease stage classification. To robustly demonstrate the utility of the generated data, the authors should evaluate NOHA in a broader range of clinical applications, such as longitudinal survival analysis or clustering.
* W2: The actual effective sample sizes for ADNI and OASIS (especially the critical "trajectories with transitions") are relatively small. Neural ODEs and generative architectures possess high parametric complexity; in such small-data regimes, there is a high risk that the model memorizes specific training trajectories rather than learning the underlying latent manifold. The manuscript lacks a detailed discussion on regularization strategies or few-shot learning designs specifically tailored to prevent such overfitting in sparse transition cohorts.
* W3: Reporting results based on only three experimental replicates (**Table 1, 2**) is statistically insufficient for validating the stability of deep generative models, which are notoriously sensitive to seed initialization. To exclude the possibility of random performance fluctuations, the authors should provide results from at least 10 replicates, accompanied by 95% confidence intervals or formal statistical significance testing (e.g., P-values) against the strongest baselines.
* W4: The NOHA objective function involves multiple components and regularization terms (**line 603**). The manuscript does not clarify the criteria for selecting the weight coefficients between $L_{recon}, L_{hazard}, L_{cls}$. A sensitivity analysis regarding these weights would be essential to evaluate the model's robustness and the trade-off between trajectory fidelity and transition awareness.

---

> ### Author Rebuttal · Authors · 2026-03-30
>
> We sincerely thank the reviewer 5AHh for your effort and time in reviewing the manuscript and highly value our research.
>
> W1) Utilize the generated data for a broader range of tasks.
>
> A) We provide a direct quantitative evaluation of the generated data to demonstrate its general applicability. Due to the character limit of rebuttal, please find the results in Reviewer 2YgR's rebuttal (Q4 & W1).
>
> W2) Limited sample size with label transitions.
>
> A) Due to the character limit, please find our reply in the rebuttal of Reviewer 2YgR (W2).
>
> W3) More experiments to show model robustness.
>
> A) We conducted $t$-tests between NOHA and the strongest baseline (ConDOR on the Amyloid experiment), which showed four second-best results out of seven metrics. As a result, our method showed statistically significant improvements in five metrics, including both accuracy for longitudinal evaluation ($p$ = 0.032 for all samples and $p$ = 0.044 for transition-only samples), and cross-sectional accuracy ($p$ = 0.010), recall ($p$ = 0.029), and specificity ($p$ = 0.003). These results demonstrate that the performance gains are not due to random variation but reflect consistent improvements across experimental runs. Also, as we acknowledge that the number of runs is limited ($n$=3), we additionally verify the stability of our method by presenting the average results from 10 replicates, along with 95% confidence intervals (CI) as follows, which show consistent performance with a tight 95% CI.
>
> | Biomarker | Methods | Longitudinal eval. |        | Cross-sectional eval. |      |         |        |             |
> |-|-|-|-|-|-|-|-|-|
> |           |         | Accuracy (All)           | Accuracy (Transition-only) | Accuracy | F1   | Precision | Recall | Specificity |
> | Cortical Thickness  | Ours (avg ± 95% CI) | 72.1 ± 3.4 | 61.7 ± 8.0 | 71.6 ± 3.2 | 71.2 ± 1.8 | 80.4 ± 4.2 | 73.3 ± 4.9 | 83.5 ± 1.9 |
> | Amyloid | Ours (avg ± 95% CI) | 61.7 ± 0.6 | 64.1 ± 3.4 | 61.6 ± 0.5 | 57.3 ± 0.6 | 72.5 ± 1.9 | 56.2 ± 1.4 | 76.1 ± 0.6 |
> | FDG | Ours (avg ± 95% CI) | 59.9 ± 0.9 | 47.4 ± 2.4 | 61.2 ± 0.9 | 60.4 ± 1.1 | 65.2 ± 1.3 | 60.8 ± 1.3 | 79.6 ± 0.6 |
> | Tau | Ours (avg ± 95% CI) | 55.4 ± 2.1 | 46.7 ± 1.9 | 56.2 ± 1.8 | 47.3 ± 1.2 | 62.8 ± 3.7 | 48.8 ± 1.3 | 75.0 ± 1.1 |
>
> W4) Trade-off of loss weights.
>
> A) We performed a sensitivity analysis of the loss weights on Amyloid using an RNN. In all settings, we set $L_{\text{ce}}=1.0$ as a reference for interpreting the effects of other losses. The results show stable performance in general and outperform baselines across most metrics.
>
> | Loss weights | | | Longitudinal eval. | | Cross-sectional eval. | | | | |
> |-|-|-|-|-|-|-|-|-|-|
> | $L_{\text{recon}}$ | $L_{\text{hazard}}$ | $L_{\text{ce}}$ |  Accuracy (All)           | Accuracy (Transition-only) | Accuracy | F1   | Precision | Recall | Specificity |
> | 2  | 1   | 1 | 60.6 ± 1.7   | 67.2 ± 2.8       | 60.4 ± 1.5       | 55.5 ± 1.6       | 66.6 ± 5.6       | 55.4 ± 2.2       | 75.8 ± 1.1       |
> | 1 | 2    | 1  | 59.2 ± 4.1 | 65.8 ± 5.0       | 58.1 ± 5.0       | 55.5 ± 3.2       | 65.3 ± 10.5      | 54.3 ± 2.4       | 74.6 ± 2.1       |
> | 0.5 | 1| 1  | 58.3 ± 1.4   | 62.1 ± 5.5       | 58.1 ± 1.6       | 52.0 ± 2.2       | 66.2 ± 4.8       | 53.2 ± 2.9       | 74.1 ± 1.4       |
> | 1 | 0.5  | 1 | 59.4 ± 0.9       | 64.1 ± 7.4       | 58.8 ± 1.1       | 53.6 ± 2.3       | 68.3 ± 4.5       | 53.7 ± 2.4       | 74.5 ± 1.3       |
>
> Q1) How to handle noisy labels?
>
> A) We agree that the dataset may include potential noise. However, since we do not explicitly know if such noises really exist and which sample is noisy, we performed a batch-wise loss update and used a normalized sampling distribution (Eq. 10) that dilute the effect of individual noise. Also, at the data preprocessing stage, we verified that all subjects with state transitions have monotonic label changes.
>
> Q2) Is the latent space invariant to confounding effects?
>
> A) We think this is a very interesting question. First, we would like to clarify that NOHA does not directly enforce strict latent invariance with respect to covariates. Instead, by explicitly inputting such factors as conditions, the latent representation focuses on modeling residual variability associated with biological progression, while conditioning variables account for known sources of variation. To investigate this, we performed two separate probing experiments by using a 3-layer MLP classifier on the learned latent representations $z_i$, where the MLP predicts binary labels of sex (M/F) and domain (ADNI/OASIS) on Tau.
>
> | Target | # Data | Label | Accuracy | Precision | Recall |
> |-|-|-|-|-|-|
> | Sex    | M:102 / F:90 | M:1 / F:0 |  50.0 | 52.9 | 85.7 |
> | Domain | ADNI:160 / OASIS:32 | ADNI:1 / OASIS:0 |79.0 | 85.3 | 90.6 |
>
> As shown in the table, the latent representation is relatively insensitive to sex but sensitive to the domain feature, likely due to inherent significant distributional differences between datasets.

---

> > ### Author Rebuttal · Reviewer_5AHh · 2026-04-01
> >
> > The authors have addressed my concerns regarding statistical significance (by providing 10-run averages with 95% CI and t-tests) and generative quality (RMSE/JSD analysis). They also clarified that the split was subject-level, which ensures the soundness of the results. While the small sample size for transition events remains a fundamental limitation of the domain, the methodological integration of Hazard-guided sampling with Neural ODEs is well-justified. I maintain my score which is already a positive result.

---

### Official Review · Reviewer_LTPj · 2026-02-26

**Soundness:** 3
**Presentation:** 3
**Significance:** 2
**Originality:** 2
**Overall Recommendation:** 2
**Confidence:** 4

**Summary:**

This paper proposes NOHA, a conditional generative model integrating Neural Ordinary Differential Equations (Neural ODE) and Continuous-Time Markov Chain (CTMC) with hazard-guided trajectory sampling, aiming to address the challenges of sparse observations and rare irreversible transitions in longitudinal disease progression modeling (e.g., Alzheimer’s disease). Experiments on four biomarkers from Alzheimer's datasets show consistent but modest improvements over nine baselines in downstream disease stage classification.

**Compliance With Llm Reviewing Policy:**

Affirmed.

**Key Questions For Authors:**

1. Provide proofs for the stability of the Neural ODE-CTMC integration and the statistical consistency of the hazard-guided sampling strategy.

2. The manuscript only considers the case where the longitudinal response is monotonic and categorical, which limits its generalizability to broader longitudinal modeling tasks.

3. Validate the model on at least one non-AD dataset (e.g., PPMI) and one non-medical longitudinal dataset to demonstrate generalizability.

4. What is the core novel contribution beyond combining existing techniques?

5. Why piecewise-constant hazards? Sensitivity analysis with continuous hazards?

6. Statistical significance testing and p-values?

7. How were baselines tuned? Are poor performances due to implementation issues?

8. Clinical expert evaluation of synthetic trajectory realism?

9. Separate results on OASIS without combining with ADNI?

10. How are L_rec and L_hazard weighted? Sensitivity to this balance?

**Limitations:**

yes.

**Strengths And Weaknesses:**

Strengths

1. This paper addresses genuine challenge of sparse, irregularly-sampled longitudinal data with severe class imbalance due to rare disease transitions.

2. This paper evaluates against nine diverse baselines (SMOTE, VAE, flow, GAN, diffusion) on four biomarkers, two datasets, and two classifiers. Includes ablation study and generation time comparison.

3. This paper achieves best transition-only accuracy (e.g., 66.7% vs. 59.3% for DDPM on cortical thickness), directly measuring ability to capture rare disease progression.

4. Synthesized trajectories exhibit expected AD patterns (amyloid accumulation in inferior parietal lobule, FDG hypometabolism in temporoparietal regions), supported by clinical literature.

Weaknesses

1. The method is exclusively validated on Alzheimer’s disease (AD) datasets with a narrow set of biomarkers. No experiments are conducted on other neurodegenerative diseases (e.g., Parkinson’s) or non-medical longitudinal data (e.g., financial risk, predictive maintenance), making it unclear whether the hazard-guided sampling and model architecture are generalizable beyond AD.


2. The paper lacks rigorous theoretical analysis of the proposed model. (i) No proof of stability or convergence for the integrated Neural ODE-CTMC framework, particularly how the CTMC hazard estimates interact with the Neural ODE’s latent dynamics. (ii) The statistical properties of the hazard-guided sampling (e.g., consistency of the augmented data distribution) are not formally established, only empirical results are provided.

3. (i) Baseline comparisons lack recently proposed generative models for longitudinal data (e.g., 2024–2025 diffusion models or transformer-based generative methods) beyond ConDOR. (ii) No analysis of the impact of key hyperparameters (e.g., Neural ODE solver type, hidden layer dimensions, number of sampling directions) on performance, leaving readers unsure of the model’s robustness to parameter choices.

4. The generated trajectories are evaluated solely via downstream classification accuracy, not via clinical validity checks (e.g., expert physician assessment of biological plausibility, consistency with known disease progression patterns). This limits the paper’s translational value for real clinical decision-making.

5.  While the paper reports generation time for a small subset (N=210), it provides no analysis of scalability to larger datasets (e.g., N>10,000) or high-dimensional data (e.g., voxel-level imaging data instead of region-wise biomarkers), which is critical for real-world deployment.

6. The ablation only compares hazard-guided vs. random sampling, but fails to isolate the contributions of individual components (e.g., CTMC vs. alternative transition models, Neural ODE vs. LSTM/Transformer for continuous dynamics).

---

> ### Author Rebuttal · Authors · 2026-03-30
>
> We sincerely thank the reviewer LTPj for your effort and time in reviewing the manuscript and for your feedback.
>
> Q1, Q6, W2) Statistical and theoretical analysis: Provide (1) a proof for the stability, (2) statistical property of the sampling (e.g., quality of augmented data), (3) a statistical significance testing with $p$-value.
>
> A) (1) The CTMC hazards do not directly alter the Neural ODE dynamics; rather, they induce a sampling distribution over subject-interval pairs, which reweights the effective training distribution. Under bounded hazards and Lipschitz continuous weight mapping, the induced sampling distribution and objective are stable with respect to perturbations in the hazard estimates, i.e., $\lVert \pi_q - \pi_{\tilde{q}} \rVert_1 \leq C \lVert q - \tilde{q} \rVert_{\infty}$, and $|L_{\pi_q}(\theta) - L_{\pi_{\tilde{q}}}(\theta)| \le C' \lVert q - \tilde{q} \rVert_{\infty}$, where $C$ and $C'$ are constant and $\pi_{q}(i,t)$ corresponds to $Pr(i,k)$ in Eq. 10. As a result, the interaction between $f_{\theta}$ and $\tilde{q}$ is stable at the objective level.
>
> (2) Due to the rebuttal character limit, please find the reply in the rebuttal to reviewer 2YgR (Q4 & W1).
>
> (3) Please find our reply in the rebuttal to reviewer 5AHh (W3).
>
> Q2, Q3, Q9, W1, W5) Additional dataset.
>
> A) Please find our reply in the rebuttal of reviewer 5out (Q2).
>
> Q4 & Q5) Novelty: Contribution beyond combining existing techniques? Why are piecewise-constant hazards used?
>
> A) First, the problem formulation itself is novel. We introduce a formulation that explicitly targets the modeling of sparse and irreversible longitudinal disease progression, opening up a new research direction not only for medical applications but also for broader domains.
>
> Second, our method uses learned transition hazards to define a sampling distribution over subject–interval pairs, which enables targeted augmentation of rare transition events.
>
> Third, our method jointly generates longitudinal biomarkers, labels, and time-varying covariates, allowing the synthesized trajectories to remain clinically consistent across multiple evolving factors.
>
> Fourth, the integration of CTMC and Neural ODE is non-trivial. Neural ODE models continuous biomarker dynamics, while CTMC captures discrete state transitions. Naively combining them would either discretize the ODE trajectory (losing the continuity advantage) or relax CTMC's Markov structure (losing interpretable hazard estimates). Our method bridges this gap using piecewise-constant hazards over observation intervals, preserving continuous-time dynamics while enabling interpretable transition modeling. This coupling goes beyond a simple combination of existing techniques.
>
> Q7) Baseline tuning?
>
> A) We performed a grid search to set learning rates in {0.0005, 0.001, 0.01, 0.1}.
>
> Q8 & W4) Clinical validity checks?
>
> A) Due to the blind review policy, we cannot disclose the involvement of clinical experts in our author list, as this may harm anonymity. Instead, we provide interpretable analyses (Sec. 4.4.2, Fig. 3), where generated trajectories align with established pathological patterns in the literature (Collij et al., Kato et al., Ossenkoppele et al., Sawyer & Kuo, Rehak et al.), studied by clinical experts.
>
> Q10) Trade-off of loss weights?
>
> A) Please find the reply in the rebuttal to reviewer 5AHh (W4).
>
> W3) Additional hyperparameter analysis and baseline comparisons.
>
> A) We analyze the impact of the Neural ODE solver and hidden dimensions on Amyloid using an RNN. To reduce the hyperparameter search space, we used the first dimension of $f_{\theta}$ to jointly define all dimensions of the encoder, decoder, $f_{\theta}$, and $g_{\phi}$. Regarding baselines, we will review the latest literature to identify additional relevant baselines. We also appreciate any specific suggestions from the reviewer.
>
> |Solver|Hidden dimension|Longitudinal eval.||Cross-sectional eval.|||||
> |-|-|-|-|-|-|-|-|-|
> |||Acc.(All)|Acc.(Transition-only)|Acc.|F1|Precision|Recall|Specificity|
> |Midpoint|64|60.0±0.4|66.8±4.6|59.2±0.5|54.8±1.6|69.2±5.3|53.8±1.4|74.7±0.3|
> |Euler|64|60.0±0.3|72.2±1.1|60.0±0.4|54.2±1.2|69.1±4.9|54.2±1.5|75.2±0.2|
> |Heun|64|61.9±0.9|69.2±3.3|61.8±0.9|57.1±1.1|68.9±5.1|57.1±2.4|76.7±1.0|
> |Heun|32|59.7±1.6|64.1±8.0|58.9±1.2|54.5±1.2|67.6±2.5|53.3±1.9|74.6±1.0|
> |Heun|16|57.0±1.0|63.1±3.7|56.7±1.8|50.9±1.0|65.3±4.0|56.7±9.2|73.3±0.9|
> |Heun|8|60.1±1.5|61.5±1.9|60.0±1.6|54.6±2.5|66.6±2.9|54.9±1.4|75.5±1.0|
>
> W6) Alternatives to CTMC and Neural ODE?
>
> A) The design choices are not interchangeable, but uniquely suited to our problem. LSTM, Transformer, and DTMC assume fixed time steps and require discretization of the timeline, making them not directly applicable for irregular time points. Moreover, no alternative transition model produces per-interval hazard estimates in a form directly usable by our sampling strategy, meaning that replacing CTMC would require redesigning the core sampling mechanism simultaneously.

---

### Official Review · Reviewer_2YgR · 2026-03-10

**Soundness:** 3
**Presentation:** 2
**Significance:** 2
**Originality:** 2
**Overall Recommendation:** 5
**Confidence:** 4

**Summary:**

This paper proposes NOHA, a conditional longitudinal generative model for sparse, irregular medical trajectories with rare and irreversible disease-stage transitions. The method combines a Neural ODE for continuous-time latent dynamics with a CTMC-style hazard module that estimates interval-wise transition risk, then uses those hazards to sample subject-interval pairs that are more likely to contain clinically meaningful progression events. The synthesized trajectories are added to the original training data to improve downstream longitudinal disease-stage prediction. Experiments are conducted on four Alzheimer’s-related biomarkers from ADNI and OASIS, with comparisons against nine baselines and evaluations using both RNN and Transformer classifiers.

**Compliance With Llm Reviewing Policy:**

Affirmed.

**Final Justification:**

All concerns addressed, this is a good contribution

**Key Questions For Authors:**

1. Was the 80/20 train-test split performed strictly at the subject level for every biomarker experiment? Relatedly, were the generator, hazard-guided sampler, and downstream classifier all trained strictly within the training fold, with no visits from test subjects used at any stage of augmentation or model selection?

2. How was hyperparameter selection performed? The appendix mentions grid search, but I did not find a clear description of the validation protocol. Was model selection confined to a validation split drawn only from the training subjects?

3. How much of the observed gain comes from hazard-guided reweighting itself versus the full Neural ODE-based generative model? A stronger disentangling ablation would help clarify where the improvements come from.

4. Can the authors provide more direct evidence of generative quality, such as trajectory-level distribution matching, transition-time calibration, nearest-neighbor or memorization analysis, and diversity metrics?

This is a promising paper with a well-motivated problem, a sensible method for irregular longitudinal disease modeling, and encouraging empirical results. The hazard-guided sampling idea is intuitive and appears to be the main reason the method works. However, I am not fully convinced yet by the generative-model claims because the validation focuses much more on downstream classification utility than on direct evidence of sample realism, diversity, and calibration. In addition, the experimental protocol needs to be stated more clearly for a longitudinal setting, especially regarding subject-level splitting and model-selection procedure. The small number of transition subjects in several experiments also makes the robustness of the conclusions harder to judge. If the split and training protocol are indeed clean and can be clarified convincingly, I would view the paper more favorably.

**Limitations:**

Yes

**Strengths And Weaknesses:**

Strengths.

1. The paper addresses an important and realistic problem setting: longitudinal clinical data are sparse, irregularly sampled, and often dominated by non-transition cases, while the clinically important state changes are rare. The motivation is strong and well aligned with the proposed method.

2. The method is conceptually appealing. Using a Neural ODE for irregular continuous-time trajectories and a CTMC-inspired hazard model for irreversible stage transitions is a sensible combination for this domain. The hazard-guided sampling mechanism is the most compelling part of the paper because it directly targets underrepresented progression intervals rather than treating all windows uniformly.

3. The empirical comparison is broader than average. The authors compare against no augmentation plus nine baselines spanning interpolation, VAE, flow, GAN, and diffusion families, and they test the augmented data with two downstream classifiers. The ablation comparing random versus hazard-guided sampling is especially helpful because it suggests the gains are not just due to adding more synthetic samples.

4. The method also appears efficient in the reported runtime experiment: in the amyloid setting, NOHA generates 210 sequences in 0.491 seconds, much faster than most deep generative baselines listed in Table 3.

Weaknesses

1. The paper’s central claim is about progression-aware longitudinal generation, but the evaluation mainly establishes downstream utility rather than generative validity. The evidence for sample quality is largely indirect: classifier performance and qualitative visualizations. I would expect stronger direct generative evaluations, such as distribution matching over trajectories, calibration of transition timing, diversity analysis, or memorization checks. The current evidence does not fully establish that the synthetic data are realistic and diverse beyond helping a downstream predictor.

2. Several experiments are extremely data-limited in exactly the cases the method emphasizes. For example, ADNI Tau has only 6 transition subjects, cortical thickness has 16, and OASIS Tau has only 2 transition subjects. Under these conditions, the reported gains are encouraging but also fragile, and the current protocol of a single 80/20 split with three random initializations feels insufficient to assess robustness. Repeated subject-level splits or bootstrap confidence intervals would make the empirical claims much more convincing.

3. The evaluation protocol is not described clearly enough for a longitudinal setting. My concern is not that the training data are used twice per se: learning a generator on the training fold and then training a downstream classifier on real plus synthetic training data is a standard augmentation setup. The issue is that, in longitudinal data, the independent unit should be the subject, not the visit or interval. Since NOHA samples subject-interval pairs and generates synthetic trajectories that are added back to the training set, any split below the subject level could allow information from the same individual to influence both the augmented training set and the test set. Appendix C states that the authors use a stratified 80/20 train-test split, but it does not explicitly say that this split is performed at the subject level. Because the evaluation is trajectory-based, this point is important to clarify, as otherwise downstream performance may partly reflect within-subject leakage rather than generalization to unseen patients.

4. The presentation around training is somewhat unclear. The paper says the synthesized trajectories are exploited in an “end-to-end manner,” and the appendix pseudo-code includes a joint objective with classification loss, but the main text reads more like a generate-then-train pipeline in which synthetic trajectories are produced and then used for downstream classification. This should be clarified carefully because it affects how readers interpret the methodological novelty and optimization procedure.

---

> ### Author Rebuttal · Authors · 2026-03-30
>
> We sincerely thank the reviewer 2YgR for your effort and time in reviewing the manuscript and highly value our research.
>
> Q1 & W3) Was the train/test sample split performed on the subject-level?
>
> A) Yes, the samples are split at the subject level. None of the complete or partial $\textit{training}$ trajectories were used at the $\textit{test}$ phase, and none of the complete or partial $\textit{test}$ trajectories were used in the $\textit{training}$. We will clearly state this setup in the revised paper.
>
> Q2) Was the hyperparameter search confined to a validation set drawn from the training set?
>
> A) Yes, the total train: val: test ratio is 7:1:2.
>
> Q3) Full Neural ODE-based generation vs. Neural ODE + hazard-guided sampling?
>
> A) We thank the reviewer for the question. This comparison is provided in Table 4, where both 'random sampling’ and 'hazard-guided sampling’ utilize Neural ODE. Specifically, the 'random sampling’ setting corresponds to the Neural ODE-based generation with random selection of subjects and time points $(i, k)$. We acknowledge that this may cause confusion and may lead to the interpretation that the 'random sampling’ setting directly uses raw data trajectories at random $(i, k)$. In the revised paper, we will explicitly state that this setting also uses NeuralODE, so that Table 4 shows the isolated effect of the hazard-guided sampling under a consistent model.
>
> Q4 & W1) Direct evaluation of generation quality.
>
> A) We thank the reviewer for this valuable suggestion. We evaluated the distributional similarity between the generated data and test data using root mean squared error (RMSE) and Jensen-Shannon Divergence (JSD) on the Tau biomarker as follows.
>
> | Method     | RMSE  $\downarrow$         | JSD    $\downarrow$          |
> |------------|----------------|----------------|
> | SMOTE      | 0.68 ± 0.03    | 0.03 ± 0.00    |
> | TVAE       | 0.60 ± 0.04    | 0.03 ± 0.01    |
> | GOGGLE     | 0.54 ± 0.01| 0.02 ± 0.00|
> | CRow       | 1.03 ± 0.08    | 0.08 ± 0.03    |
> | TimeGAN    | 1.54 ± 0.01    | 0.02 ± 0.00    |
> | CTAB-GAN   | 0.93 ± 0.02    | 0.16 ± 0.01    |
> | CTAB-GAN+  | 0.63 ± 0.02    | 0.04 ± 0.01    |
> | DDPM       | 3.71 ± 0.91    | 0.04 ± 0.02    |
> | ConDOR     | 2.15 ± 0.37    | 0.10 ± 0.01    |
> | NOHA       | 0.74 ± 0.04    | 0.03 ± 0.00    |
>
> Notably, while baseline methods generate trajectories only at the observed time points of the test data, our method generates trajectories at both observed and unobserved time points by sampling time-varying covariates. Despite this more challenging setting, our method achieves competitive distributional alignment with the test data, which demonstrates its strong potential for generalization and applicability to diverse downstream tasks.
>
> W2) Additional experiments with different subject-level splits on data-limited settings.
>
> A) We thank the reviewer for pointing out this issue. We had the same concern at the experimental design phase, and checked model performance with different data splits by resampling the subjects. Experimental results with different train and test data splits are reported below.
>
> $\bullet$ cortical thickness
>
> | Methods                         | Longitudinal evaluation |        | Cross-sectional evaluation |      |         |        |             |
> |---------------------------------|--------------------------|--------|-----------------------------|------|---------|--------|-------------|
> |                                 | Accuracy (All)          | Accuracy (Transition-only) | Accuracy | F1   | Precision | Recall | Specificity |
> |NOHA (paper) | 73.0 ± 2.8 | 66.7 ± 0.0 | 71.9 ± 2.3 | 73.1 ± 1.0 | 82.7 ± 6.5 | 75.0 ± 7.0 | 83.8 ± 1.9 |
> | NOHA (w/ different data split) | 70.6 ± 1.4 | 72.2 ± 9.6 | 69.9 ± 1.1 | 73.4 ± 1.3 | 78.3 ± 0.1 | 80.2 ± 0.6 | 82.8 ± 0.7 |
>
> $\bullet$ Tau
>
> | Methods                         | Longitudinal evaluation |        | Cross-sectional evaluation |      |         |        |             |
> |---------------------------------|--------------------------|--------|-----------------------------|------|---------|--------|-------------|
> |                                 | Accuracy (All)          | Accuracy (Transition-only) | Accuracy | F1   | Precision | Recall | Specificity |
> |NOHA (paper) | 64.9 ± 1.9      | 51.4 ± 4.8     | 61.6 ± 4.5      | 49.0 ± 1.6      | 55.4 ± 13.4     | 51.4 ± 0.2     | 78.3 ± 3.1      |
> | NOHA (w/ different data split)                    | 62.4 ± 7.1      | 51.4 ± 4.8     | 63.4 ± 6.2      | 49.5 ± 0.8      | 54.7 ± 13.9      | 52.2 ± 1.3      | 79.3 ± 3.9      |
>
> W4) Clarify the loss setup.
>
> A) We apologize for the confusion. To clarify, the model is trained in a fully end-to-end manner, where all objectives are optimized jointly. We will explicitly state the combined loss $\mathcal{L}=L_{\text{recon}} + L_{\text{hazard}} + L_{\text{cls}}$ in the method section, ensuring consistency with the pseudo-code.

---

> > ### Author Rebuttal · Reviewer_2YgR · 2026-04-03
> >
> > Thank you, i would be happy to increase my score to 5.

---

### Official Review · Reviewer_5out · 2026-03-13

**Soundness:** 3
**Presentation:** 4
**Significance:** 3
**Originality:** 3
**Overall Recommendation:** 4
**Confidence:** 2

**Summary:**

The paper proposes a conditional generative model for longitudinal disease trajectories that combines a Neural ODE with a continuous-time Markov chain to jointly model smooth biomarker dynamics and discrete, irreversible disease state transitions. A hazard network estimates piecewise-constant transition rates over each observation interval, which are converted into transition probabilities and used to preferentially sample subject-interval pairs where disease progression is likely during data synthesis. The model jointly generates biomarker trajectories, time-varying covariates, and diagnostic labels in a clinically consistent manner, and the augmented data is used to train a downstream longitudinal classifier end-to-end. The method is evaluated on four AD-related biomarkers from two neuroimaging datasets against nine baseline generative methods, consistently achieving the best or second-best longitudinal classification performance, with particularly pronounced gains in transition-only trajectory accuracy

**Compliance With Llm Reviewing Policy:**

Affirmed.

**Final Justification:**

I maintain my initial score. I think the paper overall has interesting ideas being applied to a real and significant problem area. I think other reviewers raised valid points though, hence not raising my score further in spite of the positive reply I received from the authors.

**Key Questions For Authors:**

1. The transition probabilities used for sampling are estimated by a model that is itself being trained. How stable is this sampling distribution early in training when the hazard network is poorly calibrated? Is there a warmup period or any mechanism to handle this?
2. The monotonic transition constraint is well motivated for AD but limits applicability. Are there other neurodegenerative diseases the authors considered evaluating on, or any non-medical domains where the method could be demonstrated?
3. Likewise, how necessary is the monotonic transition in practice? Could it be relaxed to allow for disease regression?

**Limitations:**

yes

**Strengths And Weaknesses:**

Soundness:
+ The main technical contribution of combining a CTMC hazard estimator with a Neural ODE for trajectory synthesis is well motivated and correctly formulated.
+ I found the ablation in Table 4 to be particularly important as it cleanly isolated the contribution of hazard-guided sampling over random sampling.
- The end-to-end joint training of the generative model and downstream classifier makes it difficult to attribute performance gains solely to generation quality, as improvements may partly reflect co-training dynamics rather than trajectory fidelity.

Presentation:
+ The paper is clearly written and the overall narrative is easy to follow. The problem motivation is compelling and well articulated.
+ Figure 1 effectively communicates the overall architecture and the relationship between the model components.
+ The pseudo-code in Appendix A is a useful addition and aids reproducibility.

Significance:
+ Longitudinal disease progression modeling under data scarcity and class imbalance is a practically important problem, and the hazard-guided sampling idea is a principled and transferable contribution beyond the AD setting.
+ The joint generation of biomarkers, labels, and time-varying covariates is a meaningful practical improvement over existing generative approaches that treat these as static conditions.
- The authors themselves note in the limitations that evaluation is restricted to AD datasets, which somewhat limits the demonstrated scope of impact.

Originality:
+ The combination of CTMC-based hazard estimation with Neural ODEs for longitudinal data synthesis is a novel and well-reasoned pairing, with the reasoning behind the combination clearly articulated.
+ The hazard-guided sampling strategy is the most original element of the work and represents a genuine conceptual contribution beyond simply applying existing generative architectures to medical data.
- The individual components (Neural ODEs for longitudinal data, CTMC for disease progression, survival-inspired hazard modeling) are each well established, and the novelty lies primarily in their integration and the sampling strategy rather than in any single component.

---

> ### Author Rebuttal · Authors · 2026-03-30
>
> We sincerely thank the reviewer 5out for your effort and time in reviewing the manuscript and highly value our research.
>
> Q1) How to handle the stability of the hazard network along epochs?
>
> A) As the reviewer pointed out, the predicted transition probabilities and the hazard network are unstable in the early training stages. To handle this issue, we adopt a gradual stabilization strategy by using a LambdaLR learning rate scheduler with a decay rate of 0.995 per epoch. By gradually reducing the learning rate, we observe that fluctuations in all losses and metrics are substantially reduced over epochs, which suggests that it helps stabilize the optimization of the network.
>
> Q2) Are there other possible applicative domains?
>
> A) Yes, during experimental planning, we considered extending our evaluation on the Parkinson’s Progression Markers Initiative (PPMI) dataset, which is also a neurodegenerative disease dataset with three classes (CN, Prodromal, PD). We investigated all available longitudinal samples with MRI (T1, T1-weighted, and T2), but found none of them had a label change over time. Without any label transitions, it is infeasible to measure a transition hazard, and therefore, the experiment falls out of the scope of this work’s motivation (i.e., capturing significant state transitions). Moreover, we also considered solely using the OASIS dataset, but it contains only $\textit{one}$ sample that has a label transition, which makes model training infeasible without integrating the ADNI subjects. We expect to utilize such additional datasets in future work if sufficient longitudinal samples with label changes become available. Beyond medical applications, we expect that our framework can also be applicable to non-medical domains characterized by monotonic state transitions, such as battery aging processes and the physical degradation and fatigue of materials.
>
> Q3) The necessity of monotonic transition in practice?
>
> A) We thank the reviewer for this important question. In our framework, the monotonic transition constraint is implemented through the structure of the $C$-state continuous-time Markov chain (CTMC), where transitions are only allowed from state $c$ to $c+1$. This design is motivated by the clinical characteristics of neurodegenerative diseases, in which disease progression is irreversible at the pathological level.
> We acknowledge that regression may very rarely occur in practice due to diagnostic uncertainty and potential confounding factors. However, these cases are extremely rare, and it is more likely to reflect transient variability in observations rather than true reversal of the underlying disease process. To incorporate these reversible cases, a new model architecture is necessary, where the CTMC is replaced with a different structure that allows backward transitions.

---

> > ### Author Rebuttal · Reviewer_5out · 2026-04-02
> >
> > I thank the authors for their response. They addressed my main points from my review.

---

### Decision · Program_Chairs · 2026-04-30

**Decision:**

Reject

**Comment:**

After the author/reviewer discussion, 3 out of the 4 reviewers leaned toward acceptance and the 1 reviewer favoring rejection did not respond to the author rebuttal. From carefully reading through the discussion, I think that the authors have proposed a nontrivial novel neural ODE-based generator of disease trajectories that addresses the interesting situation of irreversible transitions. Unfortunately, this setting appears a bit niche as there don't seem to be many publicly available datasets that have enough of these irreversible transitions. I'm inclined to suggest that the authors try to find more datasets to demonstrate their proposed approach on, possibly even ones that are synthetic and/or semi-synthetic (although these wouldn't be as compelling as real data examples). Doing so would substantially strengthen the paper.